# GRAPH GENERATION WITH DESTINATION-PREDICTING DIFFUSION MIXTURE

## ABSTRACT

Generation of graphs is a major challenge for real-world tasks that require under-standing the complex nature of their non-Euclidean structures. Although diffusion models have achieved notable success in graph generation recently, they are ill-suited for modeling the structural information of graphs since learning to denoise the noisy samples does not explicitly capture the graph topology. To tackle this limitation, we propose a novel generative framework that models the topology of graphs by predicting the destination of the diffusion process, which is the original graph that has the correct topology information, as a weighted mean of data. Specifically, we design the generative process as a mixture of diffusion processes conditioned on the endpoint in the data distribution, which drives the process toward the predicted destination, resulting in rapid convergence. We introduce new simulation-free training objectives for predicting the destination, and further discuss the advantages of our framework that can explicitly model the graph topology and exploit the inductive bias of the data. Through extensive experimental validation on general graph and 2D/3D molecule generation tasks, we show that our method outperforms previous generative models, generating graphs with correct topology with both continuous (e.g. 3D coordinates) and discrete (e.g. atom types) features.

## 1 INTRODUCTION

Generation of graph-structured data has emerged as a crucial task for real-world problems such as drug discovery (Simonovsky & Komodakis, 2018), protein design (Ingraham et al., 2019), and program synthesis (Brockschmidt et al., 2019). To tackle the challenge of learning the underlying distribution of graphs, deep generative models have been proposed, including models based on generative adversarial networks (GANs) (De Cao & Kipf, 2018; Martinkus et al., 2022), recurrent neural networks (RNNs) (You et al., 2018), and variational autoencoders (VAEs) (Jin et al., 2018).

Recently, diffusion models have achieved state-of-the-art performance on the generation of graph-structured data (Niu et al., 2020; Jo et al., 2022; Hoogeboom et al., 2022). These models learn the generation process as the time reversal of the forward process, which corrupts the graphs by gradually adding noise that destroys its topological properties. Since the generative process is derived from the unknown score function (Song et al., 2021) or noise (Ho et al., 2020), existing graph diffusion models aim to estimate them in order to denoise the data from noise, which are commonly referred to as the *denoising diffusion models* (Figure 1 (a)).

Despite their success, learning the score or noise is fundamentally ill-suited for the generation of graphs. Although the key to generating valid graphs is modeling the topological information such as connectivity or clusteredness, the score or noise does not explicitly model these features. Thereby it is challenging for the diffusion models to recover the topological properties from the corrupted graphs through denoising, which leads to failure cases even for small graphs. A way to more accurately generate graphs with correct topology would be explicitly learning to predict the final graph to be generated, instead of learning how to denoise a noisy version of the original graph.

However, predicting the destination of the generative process in a deterministic manner is challenging since the prediction would be highly inaccurate in the early steps of the diffusion process, and such an inaccurate prediction may lead the process in the wrong direction, resulting in invalid results. Few existing works (Hoogeboom et al., 2021; Austin et al., 2021; Vignac et al., 2022) based on denoising diffusion models aim to predict the probability of the data by parameterizing the denoising process,

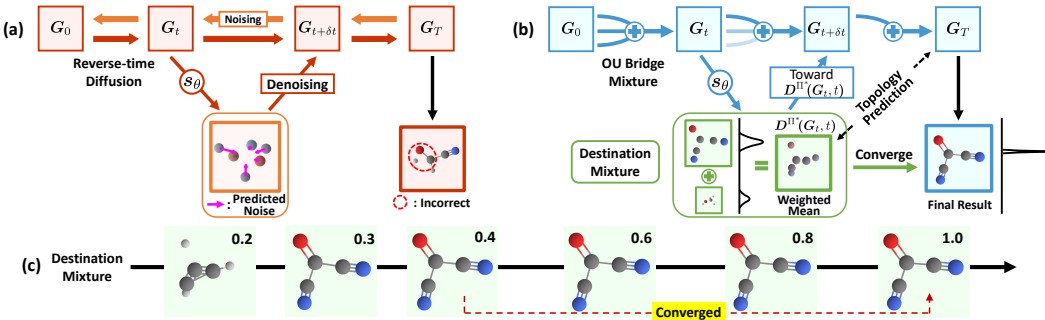

Figure 1: **Illustration of the graph generative process.** (a: Denoising diffusion model, b: DruM (ours), c: Destination mixture of DruM) Our DruM (**blue**) successfully generates graphs with valid topology by predicting the final result, via learning the destination mixture (Eq. (1)) as a weighted mean of data (**green**). The predicted destination of DruM converges in an early stage to the correct topology as visualized in (c). To this end, we design the generative process as a mixture of endpoint-conditioned diffusion processes (Eq. (2)), namely the OU bridge mixture (Eq. (4)), which is driven toward the destination mixture. On the other hand, previous denoising diffusion models (**red**) often fail to capture the correct topology as they learn the score or noise for denoising, without explicit knowledge of the destination.

but it is only applicable to categorical data with a finite number of states and thus cannot generate graphs with continuous features, which is not suitable for tasks such as 3D molecule generation.

To address these limitations of existing diffusion models, we propose a novel generative diffusion framework that explicitly models the graph topology by predicting the destination of the generative process as a weighted mean of data (Figure 1 (b)). Specifically, we design the generative process as a mixture of Ornstein-Uhlenbeck processes conditioned on the endpoint in the data distribution which is different from the denoising diffusion process, where the drift drives the process toward the predicted destination. We establish a theoretical basis for our destination-predicting generative framework (Section 3.1) and introduce new simulation-free objectives for learning to predict the destination, which is equivalent to maximizing the likelihood (Section 3.2). Thanks to its ability to predict the destination of the generative process which has an accurate topology of the graph being generated, our framework converges faster to the correct graph topology in an early sampling step (Figure 1 (c) and Figure 3 (Right)).

We experimentally validate our method on diverse real-world graph generation tasks. We first validate it on general graph generation benchmarks with synthetic and real-world graphs, on which it outperforms previous deep graph generative models including graph diffusion models, by being able to generate valid graphs with correct topologies. We further validate our method on 2D and 3D molecule generation tasks to demonstrate its ability to generate graphs with both the continuous and discrete features, on which ours generates a significantly larger number of valid and stable molecules compared to the state-of-the-art baselines. Our main contributions can be summarized as follows:

- We observe that previous graph diffusion models cannot accurately model the graph topology as they learn to denoise at each step w/o the knowledge of the destination of the generative process.
- To fix such a myopic behavior of previous diffusion models, we propose a novel graph generation framework that captures the graph topology by directly predicting the destination of the generative process modeled by a mixture of endpoint-conditioned diffusion processes.
- We derive theoretical groundwork for destination-predicting framework and discuss its advantages.
- Our method significantly outperforms previous graph diffusion models on the generation of diverse real and synthetic graphs, as well as on 2D/3D molecule generation tasks, by being able to generate graphs with accurate topologies, and both the discrete and continuous features.

## 2 RELATED WORK

**Diffusion models**   Diffusion models have been shown to successfully generate high-quality samples from diverse data domains such as images (Dhariwal & Nichol, 2021; Saharia et al., 2022), audios (Chen et al., 2021; Jeong et al., 2021), point clouds (Cai et al., 2020), and videos (Ho et al., 2022). Despite their success, existing diffusion models for graphs (Niu et al., 2020; Jo et al., 2022) often fail to generate graphs with correct structures since the denoising process they use does not explicitly consider the graph topology. Discrete diffusion model (Vignac et al., 2022) proposed

to model the noising process as successive graph edits for graphs with categorical node and edge attributes, but this is not a desirable solution for real-world graph generation tasks since it cannot be alone applied to graphs with continuous features, such as the 3D coordinates of atoms. To address these limitations, we propose a novel graph diffusion framework that models the generative process as a mixture of diffusion processes and learns to predict the destination as a weighted mean of data, instead of predicting the denoising function at each step. This promotes our generative process to be driven toward the predicted destination, resulting in generating valid graphs with correct topology.

**Diffusion bridge process** A line of recent works has improved the generative framework of diffusion models by leveraging the diffusion bridge processes, which are processes conditioned to an endpoint. Schrödinger Bridge (Vargas et al., 2021; Bortoli et al., 2021b; Chen et al., 2022; Shi et al., 2023) aims to find both the forward and the backward process that transforms arbitrary distributions back and forth using iterative proportional fittings that require heavy computations. More recent works (Peluchetti, 2021; Wu et al., 2022; Ye et al., 2022; Liu et al., 2023) consider learning the generation process as a mixture of diffusion processes instead of reversing the noising process as in denoising diffusion models. Peluchetti (2021) introduces a diffusion mixture representation that constructs a generation process as a mixture of the bridge processes. Wu et al. (2022) injects physical information into the process by adding informative prior to the drift, while Ye et al. (2022) and Liu et al. (2022) extend the bridge process to constrained domains. However, approximating the drift as in these works cannot accurately capture the discrete structure of graphs since it does not explicitly consider the topology, and could be problematic as the drift diverges near the terminal time. Instead, we propose a new approach to learning to predict the destination of the generative process by leveraging the diffusion mixture, which allows it to model the graph topology.

**Graph generative models** Deep generative models for graphs either generate nodes and edges in an autoregressive manner using VAE (Jin et al., 2018), RNN (You et al., 2018), normalizing flow (Luo et al., 2021; Shi et al., 2020), and attention (Liao et al., 2019), or generate all the nodes and edges at once using GAN (De Cao & Kipf, 2018; Martinkus et al., 2022), VAE (Ma et al., 2018), or normalizing flow (Madhawa et al., 2019; Zang & Wang, 2020). However, these models show poor performance due to restrictive model architectures for modeling the likelihood or their inability to model the permutation equivariant nature of graphs. Recently, diffusion models for graphs (Niu et al., 2020; Jo et al., 2022; Vignac et al., 2022; Hoogeboom et al., 2022) have made large progress in generating synthetic graphs as well as molecules. However, existing graph diffusion models either fail to capture the graph topology or are not applicable to general tasks due to the structural restriction of the framework. To overcome these limitations, we propose a novel graph diffusion framework that explicitly models the topology by learning to predict the destination. Our method largely outperforms existing models (Jo et al., 2022; Vignac et al., 2022; Hoogeboom et al., 2022) on general graph generation as well as 2D and 3D molecule generation tasks.

## 3 DESTINATION-PREDICTING DIFFUSION MIXTURE FOR GRAPH GENERATION

In this section, we present our novel graph generative framework, **D**estination-**Pr**edicting Diff**u**sion **M**ixture (DruM), which learns to predict the destination using the mixture of diffusion processes. Throughout the paper, a graph $G$ with $N$ nodes is defined by a pair $(X, A)$ where $X \in \mathbb{R}^{N \times F}$ is the node features with feature dimension $F$ and $A \in \mathbb{R}^{N \times N}$ is the weighted adjacency matrix

### 3.1 DESIGNING THE GRAPH GENERATIVE PROCESS

The key to generating graph-structured data is understanding the underlying topology of graphs which is crucial to determining its validity, since a slight modification in the edges may significantly change its structure and the attributes, for example, planarity or the molecular properties. However, previous diffusion models fail to do so as their objective is to denoise the noisy graphs, in which the topology is only implicitly captured (Figure 1 (a)). To overcome the limitation, we propose to design a generative framework that can directly learn to predict the accurate graph structures and their structural properties for capturing valid topology.

**Destination mixture** Our goal is to directly predict the destination of the diffusion process that transports the prior distribution to the data distribution $\Pi^*$. To be specific, for a diffusion process represented as a trajectory of random variables $\{G_\tau\}_{\tau \in [0,T]}$, we aim to predict the terminus of the process $G_T$ given the current state $G_t$. However, identifying the exact destination at the early stage

of the process is problematic, since $G_t$ contains almost no information. Hence predicting a single deterministic data could lead the generative process in the wrong direction.

To address this problem, we present a new approach to predicting the *probable destination*, which we define as a weighted mean of the destinations (Figure 1 (b)). Since the probability of a graph $g$ being the destination is equal to the transition probability of the process denoted as $p_{T|t}(g|\cdot)$, we define the probable destination given the current state $G_t$ via the expectation as follows:

$$D(G_t, t) = \int g \cdot p_{T|t}(g|G_t) \, \mathrm{d}g, \tag{1}$$

which we refer to as the *destination mixture* of the process, visualized in Figure 1 as green. In order to explicitly model this, we construct a generative process as a mixture of diffusion processes that is driven toward the destination mixture, which we describe in the following paragraphs.

**Ornstein-Uhlenbeck bridge process**     As a building block of our generative framework, we leverage diffusion processes with fixed endpoints, namely the *diffusion bridge* processes. A bridge process that is destined to end up on a fixed point can be derived from a reference process $\mathbb{Q} : \mathrm{d}G_t = f(G_t, t)\mathrm{d}t + \sigma_t\mathrm{d}W_t$, to be conditioned to an endpoint $g$ by applying the Doob's h-transform (Doob & Doob, 1984) as follows:

$$\mathrm{d}G_t = \left[ f(G_t, t) + \sigma_t^2 \nabla_{G_t} \log p_{T|t}(g|G_t) \right] \mathrm{d}t + \sigma_t \mathrm{d}W_t.$$

We propose to use a new family of bridge processes, namely the *Ornstein-Uhlenbeck* (OU) bridge process that provides flexibility for designing the complex generative process. (see Section A.1 of the Appendix for the derivation):

$$\mathbb{Q}^g \; : \; \mathrm{d}G_t = \left[ \underbrace{\alpha\sigma_t^2 G_t + \frac{\sigma_t^2}{v_t}\left( \frac{g}{u_t} - G_t \right)}_{\eta^g(G_t, t)} \right]\mathrm{d}t + \sigma_t\mathrm{d}W_t, \quad G_0 \sim p_{prior}, \tag{2}$$

where $\alpha$ is a constant, $\sigma_t$ is a scalar function, and $W_t$ is the standard Wiener process, $p_{prior}$ is a prior distribution, and the scalar functions $u_t$ and $v_t$ are defined as follows:

$$u_t = \exp\left( \alpha \int_t^T \sigma_\tau^2 \mathrm{d}\tau \right), \; v_t = \frac{1}{2\alpha}\left( 1 - u_t^{-2} \right). \tag{3}$$

The destination of this process is fixed to $G_T = g$, since the drift $\eta^g(\cdot, t)$ of the process in Eq. (2) forces the trajectory $G_t$ towards the destination $g$. Although there exists a more general class of bridge processes with non-linear drift (see Section A.1 of the Appendix), they have intractable transition probability and require expensive SDE simulation to obtain trajectories. In contrast, the OU bridge process yields tractable transition probabilities due to its affine nature that allows the training of our generative model to be simulation-free, which we further discuss in Section 3.2. Note that the Brownian bridge process used in previous works (Wu et al., 2022; Liu et al., 2022) is a special case of the OU bridge process when $\alpha \to 0$ (see Section A.1 of the Appendix). With the OU bridge processes in hand, we present a generative process for direct prediction of the destination.

**Diffusion mixture for destination prediction**     As the destination mixture in Eq. (1) is a weighted mean of endpoints, conceptually, this can be modeled by aggregating the endpoint-conditioned processes with respect to these weights. Inspired by the diffusion mixture (Peluchetti, 2021; Wu et al., 2022; Liu et al., 2022), we design the generation process by mixing the OU bridge processes conditioned on the endpoints from the data distribution. Here, we leverage the *diffusion mixture representation* (Brigo, 2008; Peluchetti, 2021), which yields the representation of a mixture process that combines a collection of diffusion processes. In a nutshell, the SDE representation of the mixture process is modeled by the weighted mean of the SDEs of the diffusion process in the collection (we provide a full definition in Section A.2 of the Appendix).

To be more precise, we mix a collection of OU bridge processes $\{\mathbb{Q}^g : g \sim \Pi^*\}$ to construct a generative process, for which the mixture process is modeled by the following SDE:

$$\mathbb{Q}^{\Pi^*} \; : \; \mathrm{d}G_t = \left[ \int \eta^g(G_t, t)\frac{p_t^g(G_t)}{p_t(G_t)}\Pi^*(\mathrm{d}g) \right]\mathrm{d}t + \sigma_t\mathrm{d}W_t, \quad G_0 \sim p_{prior}, \tag{4}$$

where $p_t^{\boldsymbol{g}}$ is the marginal density of the bridge process $\mathbb{Q}^{\boldsymbol{g}}$ and $p_t(\cdot) := \int p_t^{\boldsymbol{g}}(\cdot)\Pi^*(\mathrm{d}\boldsymbol{g})$ is the marginal density of the mixture process. Notably, the terminal distribution of the mixture process $\mathbb{Q}^{\Pi^*}$ is equal to the data distribution $\Pi^*$ by construction. We refer to this mixture process as the *OU bridge mixture*.

Remarkably, the mixture process $\mathbb{Q}^{\Pi^*}$ can be explicitly represented in terms of the destination mixture. To be specific, the drift of $\mathbb{Q}^{\Pi^*}$ can be derived from the SDE representation of the OU bridge process in Eq. (2) as follows (see Section A.3 of the Appendix for the derivation of the drift):

$$\eta(\boldsymbol{G}_t, t) = \alpha\sigma_t^2\boldsymbol{G}_t + \frac{\sigma_t^2}{v_t}\left(\frac{1}{u_t}\boldsymbol{D}^{\Pi^*}(\boldsymbol{G}_t, t) - \boldsymbol{G}_t\right), \quad \boldsymbol{D}^{\Pi^*}(z, t) := \int \boldsymbol{g}\frac{p_t^{\boldsymbol{g}}(z)}{p_t(z)}\Pi^*(\mathrm{d}\boldsymbol{g}). \quad (5)$$

Notice that from the definition of the transition distribution, we can derive the following:

$$\boldsymbol{D}^{\Pi^*}(\boldsymbol{G}_t, t) = \int \boldsymbol{g}\frac{p_t^{\boldsymbol{g}}(\boldsymbol{G}_t)}{p_t(\boldsymbol{G}_t)}\Pi^*(\mathrm{d}\boldsymbol{g}) = \int \boldsymbol{g}\frac{p(\boldsymbol{G}_t|\boldsymbol{G}_T=\boldsymbol{g})p(\boldsymbol{G}_T=\boldsymbol{g})}{p_t(\boldsymbol{G}_t)}\mathrm{d}\boldsymbol{g} = \int \boldsymbol{g}\, p_{T|t}(\boldsymbol{g}|\boldsymbol{G}_t)\mathrm{d}\boldsymbol{g}, \quad (6)$$

which shows that $\boldsymbol{D}^{\Pi^*}(\cdot, t)$ coincides with the destination mixture of $\mathbb{Q}^{\Pi^*}$ in Eq. (1). As a result, $\boldsymbol{D}^{\Pi^*}(\cdot, t)$ is the prediction at time $t$ of the final graph to be generated. A key observation here is that the marginal density $p_t^{\boldsymbol{g}}$ converges to 1 if $\boldsymbol{g}$ corresponds to the final graph while the probability becomes 0 otherwise, due to the endpoint condition of the bridge processes. Thereby, the destination mixture converges to the final graph, and this convergence is achieved at an early stage as visualized in Figure 1 (c) and Section E.2 of the Appendix. We further analyze the convergence behavior with respect to the coefficient $\alpha$ and the noise schedule $\sigma_t$ in Section D.2 of the Appendix.

In particular, the drift of the OU bridge mixture in Eq. (5) highly resembles the drift of the OU bridge process in Eq. (2), except that the endpoint is replaced by the destination mixture. From this observation, we can see that the trajectory of the OU bridge mixture is driven toward the destination mixture as the drift guides the process to the direction of $\boldsymbol{D}^{\Pi^*}(\cdot, t)$, where the process terminates in the data distribution by construction. Thus, if we could estimate the destination mixture, the OU bridge mixture can be used as a generative model where the estimated destination mixture acts as the prediction of the destination through the generative process. In this sense, we name our proposed generative framework as *Destination-Predicting Diffusion Mixture (DruM)*. We further extend the framework for the generation of *attributed graphs* in Section A.4 of the Appendix, which allows us to directly model the graph topology with both the continuous and discrete features.

Before introducing new training objectives for estimating the destination mixture, we discuss the difference of our generative process from the denoising diffusion models. The generative process of DruM which is modeled by the mixture of bridge processes describes the exact transport from the prior distribution to the data distribution $\Pi^*$, whereas the time reversal of denoising diffusion models is not an exact transport to the data distribution for finite terminal time $T$, for instance, SMLD (Song & Ermon, 2019) or DDPM (Ho et al., 2020). We provide further discussion on the difference between our mixture process and the denoising diffusion processes in Section A.11.

## 3.2 LEARNING THE DESTINATION MIXTURE

In this section, we introduce new training objectives for the generative model via estimating the destination mixture and further discuss the advantages of our framework.

**Training objectives** Our goal is to explicitly model the graph topology by learning to predict the destination of the generative process. Thus we design the generative process as an OU bridge mixture and aim to estimate the destination mixture using a neural network $\boldsymbol{s}_\theta(\cdot, t)$. Remarkably, we show that estimating the destination mixture is equivalent to maximizing the likelihood.

We propose to define the generative model $\mathbb{P}^\theta$ to approximate the mixture process $\mathbb{Q}^{\Pi^*}$ as follows:

$$\mathbb{P}^\theta : \mathrm{d}\boldsymbol{G}_t = \eta_\theta(\boldsymbol{G}_t, t)\mathrm{d}t + \sigma_t\mathrm{d}\mathbf{W}_t, \quad \eta_\theta(\boldsymbol{G}_t, t) = \alpha\sigma_t^2\boldsymbol{G}_t + \frac{\sigma_t^2}{v_t}\left(\frac{1}{u_t}\boldsymbol{s}_\theta(\boldsymbol{G}_t, t) - \boldsymbol{G}_t\right), \quad (7)$$

where $\boldsymbol{s}_\theta$ is desired to estimate the destination mixture. In order to train $\mathbb{P}^\theta$ to approximate $\mathbb{Q}^{\Pi^*}$ via maximum likelihood estimation, we leverage the Girsanov theorem (Øksendal, 2003) for upper bounding the KL divergence between the data distribution $\Pi^*$ and the terminal distribution of the generative model $\mathbb{P}^\theta$ denoted as $p_T^\theta$ (see Section A.6 for the application of the Girsanov theorem):

$$D_{KL}(\Pi^*\|p_T^\theta) \le D_{KL}(\mathbb{Q}^{\Pi^*}\|\mathbb{P}^\theta) = \mathbb{E}_{\boldsymbol{G}\sim\mathbb{Q}^{\Pi^*}}\left[\frac{1}{2}\int_0^T \gamma_t^2\left\|\boldsymbol{s}_\theta(\boldsymbol{G}_t, t) - \boldsymbol{D}^{\Pi^*}(\boldsymbol{G}_t, t)\right\|^2\mathrm{d}t\right] + C, \quad (8)$$

where $\boldsymbol{G} \sim \mathbb{Q}^{\Pi^*}$ denotes the sampled trajectories from the OU bridge mixture, $\gamma_t := \sigma_t/(u_t v_t)$, and $C$ is a constant independent of $\theta$. However, since the ground truth destination mixture of $\mathbb{Q}^{\Pi^*}$ is not analytically accessible, Eq. (8) cannot be used directly. Therefore, using the definition of the destination mixture, we introduce a new tractable objective for estimating the destination mixture that is equivalent to minimizing Eq. (8) (see Section A.6 of the Appendix for the derivation):

$$\mathcal{L}(\theta) = \mathbb{E}_{\boldsymbol{G} \sim \mathbb{Q}^{\Pi^*}}\left[\frac{1}{2}\int_0^T \gamma_t^2 \|\boldsymbol{s}_\theta(\boldsymbol{G}_t, t) - \boldsymbol{G}_T\|^2 \mathrm{d}t\right]. \tag{9}$$

Note that the goal of the loss in Eq. (9) is for $\boldsymbol{s}_\theta$ to estimate the destination mixture $\boldsymbol{D}^{\Pi^*}(\boldsymbol{G}_t, t)$ not the exact endpoint $\boldsymbol{G}_T$, and we refer to this objective as the *destination mixture matching*. Learning to predict the destination not only allows us to directly model the topology of the final graph, but further guarantees the terminal distribution of our generative model to closely approximate the data distribution. We derive in Section A.10 that learning the destination mixture is not interchangeable with learning the score function of the mixture process, and further discuss the difference from the training objectives of denoising diffusion models in Section A.11.

During training, the trajectories $\boldsymbol{G} \sim \mathbb{Q}^{\Pi^*}$ can be easily obtained by first sampling $\boldsymbol{G}_0 \sim p_{prior}$, $\boldsymbol{G}_T \sim \Pi^*$ and $t \sim [0, T]$, then sampling $\boldsymbol{G}_t \sim p_{t|0,T}(\boldsymbol{G}_t|\boldsymbol{G}_0, \boldsymbol{G}_T)$ which is the distribution of $\mathbb{Q}^{\Pi^*}$ at time $t$ given the endpoints $\boldsymbol{G}_0$ and $\boldsymbol{G}_T$. We highlight that this probability is analytically computable (see Section A.7 and Section A.8 of the Appendix) due to the affine nature of the OU bridge process, and therefore training with the destination mixture matching is simulation-free. Our approach of using the transition probability is 17.5 times faster compared to Wu et al. (2022) which requires expensive SDE simulation.

**Sampling** We can generate valid graphs by starting from samples from the prior distribution and simulating the parameterized bridge process of Eq. (7) from time $t = 0$ to $t = T$, where the drift is computed from the trained model $\boldsymbol{s}_\theta$. We can leverage any SDE solver used in previous works, for example, Euler-Maruyama method or Heun's 2nd order method. Note that for the generation of attributed graphs, we generate the node features and the adjacency matrices simultaneously using the system of SDEs as in Eq. (31) which we describe in Section A.4. We provide the pseudo-code for the training and sampling in Section B.1 and further explain in details in Section B.2 and Section B.4.

**Advantages of our framework** We conclude this section by explaining the advantages of our framework. First, DruM can directly model the graph topology by predicting the destination instead of implicitly capturing via noise or score. The destination mixture matching guarantees that learning the topology of the final graph is equivalent to learning the generative model as a diffusion process that transports the prior distribution to the data distribution. Furthermore, our framework is not restricted to the type of data to be generated, since there is no constraint on data representation for the parameterization of our generative model. Thus our framework is applicable to both continuous and discrete data, for example, 3D molecules with both discrete atom types and continuous coordinates.

From the perspective of the model hypothesis space, learning the destination mixture is considerably easier compared to previous objectives such as learning the score function or the drift of the diffusion process. While the destination mixture is supported inside the bounded data space, the score function or the drift tends to diverge near the terminal time which could be problematic for the model to learn. Furthermore, we can exploit the inductive bias of the data for learning the destination mixture, which is critical as it dramatically reduces the hypothesis space. To be specific, we can leverage the prior knowledge of the data representation such as one-hot encoding or the categorical type by adding an additional function at the last layer of the model $\boldsymbol{s}_\theta$, for instance, softmax function for the one-hot encoded node features and the sigmoid function for the 0-1 adjacency matrices (see Section B.3 of the Appendix for details). We experimentally verify these advantages in Section 4.

## 4 EXPERIMENTS

### 4.1 GENERAL GRAPH GENERATION

We validate DruM on general graph generation tasks to show that it can generate valid graph topology.

**Datasets and metrics** We evaluate the quality of generated graphs on three synthetic and real datasets used as benchmarks in previous works (Martinkus et al., 2022; Vignac et al., 2022): **Planar**,

Table 1: **Generation results on the general graph datasets.** Best results are highlighted in bold, where smaller MMD and larger V.U.N. indicate better results. Baseline results are taken from Vignac et al. (2022) or obtained by running the open-source codes. Hyphen(-) denotes out-of-resources that take more than 2 weeks.

| | Planar | | | | | SBM | | | | | Proteins | | | |
| | Synthetic, $|V| = 64$ | | | | | Synthetic, $44 \leq |V| \leq 187$ | | | | | Real, $100 \leq |V| \leq 500$ | | | |
| | Deg. | Clus. | Orbit | Spec. | V.U.N. | Deg. | Clus. | Orbit | Spec. | V.U.N. | Deg. | Clus. | Orbit | Spec. |
|---|---|---|---|---|---|---|---|---|---|---|---|---|---|---|
| Training set | 0.0002 | 0.0310 | 0.0005 | 0.0052 | 100.0 | 0.0008 | 0.0332 | 0.0255 | 0.0063 | 100.0 | 0.0003 | 0.0068 | 0.0032 | 0.0009 |
| GraphRNN | 0.0049 | 0.2779 | 1.2543 | 0.0459 | 0.0 | 0.0055 | 0.0584 | 0.0785 | 0.0065 | 5.0 | 0.0040 | 0.1475 | 0.5851 | 0.0152 |
| GRAN | 0.0007 | 0.0426 | 0.0009 | 0.0075 | 0.0 | 0.0113 | 0.0553 | 0.0540 | 0.0054 | 25.0 | 0.0479 | 0.1234 | 0.3458 | 0.0125 |
| SPECTRE | 0.0005 | 0.0785 | 0.0012 | 0.0112 | 25.0 | 0.0015 | 0.0521 | **0.0412** | 0.0056 | 52.5 | 0.0056 | 0.0843 | **0.0267** | 0.0052 |
| EDP-GNN | 0.0044 | 0.3187 | 1.4986 | 0.0813 | 0.0 | 0.0011 | 0.0552 | 0.0520 | 0.0070 | 35.0 | - | - | - | - |
| GDSS | 0.0041 | 0.2676 | 0.1720 | 0.0370 | 0.0 | 0.0212 | 0.0646 | 0.0894 | 0.0128 | 5.0 | 0.0861 | 0.5111 | 0.732 | 0.0748 |
| ConGress | 0.0048 | 0.2728 | 1.2950 | 0.0418 | 0.0 | 0.0273 | 0.1029 | 0.1148 | - | 0.0 | - | - | - | - |
| DiGress | **0.0003** | 0.0372 | **0.0009** | 0.0106 | 75 | 0.0013 | 0.0498 | 0.0434 | 0.0400 | 74 | - | - | - | - |
| **DruM (Ours)** | 0.0005 | **0.0353** | **0.0009** | **0.0062** | **90.0** | **0.0007** | **0.0492** | 0.0448 | **0.0050** | **85.0** | 0.0019 | 0.0660 | 0.0345 | **0.0030** |

Figure 2: **(Left) Topology analysis** through the generative process. We compare Spec. MMD and V.U.N of the destination mixture from DruM against the implicit destinations computed from GDSS, ConGress, and DiGress which we provide details in Appendix C.1. **(Middle) MMD between the test set and the destination mixture of DruM** through the generative process. **(Right) The complexity of DruM** with and without using the inductive bias, measured by the Frobenius norm of the Jacobian of the models.

Stochastic Block Model (**SBM**), and **Proteins** (Dobson & Doig, 2003). We follow the evaluation setting of Martinkus et al. (2022) using the same data split. We measure the maximum mean discrepancy (MMD) of four graph statistics between the set of generated graphs and the test set: degree (**Deg.**), clustering coefficient (**Clus.**), count of orbits with 4 nodes (**Orbit**), and the eigenvalues of the graph Laplacian (**Spec.**). To verify that the model truly learns the distribution, we report the percentage of valid, unique, and novel (**V.U.N.**) graphs for which the validness is defined as satisfying the specific property of each dataset. We provide further details in Section C.1 of the Appendix.

**Baselines** We compare DruM against the following graph generative models: **GraphRNN** (You et al., 2018) an autoregressive model based on RNN, **GRAN** (Liao et al., 2019) an autoregressive model with attention, **SPECTRE** (Martinkus et al., 2022) a one-shot model based on GAN, **EDP-GNN** (Niu et al., 2020) a score-based model for adjacency matrix, **GDSS** (Jo et al., 2022) and **ConGress** (Vignac et al., 2022) a continuous diffusion model, and **DiGress** (Vignac et al., 2022), a discrete diffusion model. We provide the details of training and sampling of DruM in Section B of the Appendix and describe further implementation details in Section C.1 of the Appendix.

**Results** Table 1 shows that our DruM outperforms all the baselines on all datasets. Especially, DruM achieves the highest validity (V.U.N.) metric, as it accurately learns the underlying topology of the graphs. Notably, our method outperforms DiGress by a large margin in V.U.N., even though we do not use specific prior distributions or structural feature augmentation that are utilized in DiGress. We provide an ablation study on the model architecture in Section D.2 to validate that the superior performance of DruM comes from its ability to accurately model the graph topology by predicting the destination mixture. We provide the visualization of the generated graphs and the generative process of DruM in Section E, showing that DruM can accurately capture the attributes of each dataset.

**Topology analysis** To show how the destination prediction results in graphs with correct topology, we conduct an analysis of the destination mixture. Figure 2 (Left) demonstrates that DruM is able to achieve the spectral property of the target graph at an early stage by explicitly modeling the topology via predicting the destination mixture. In contrast, GDSS and ConGress fail to recover the spectral properties as they implicitly model the topology via predicting the noise or score functions. Further, ours recovers the spectral property faster than DiGress, resulting in graphs with higher validity. In particular, we observe that the V.U.N. of the estimated destination mixture increases after achieving the desired spectral property, resulting in 90% V.U.N. This shows that predicting the final graph to be generated allows us to better capture the global topologies. Moreover, we plot the MMD results

Table 2: **Generation results on the 2D molecule datasets.** We report the mean of 3 different runs. Best results are highlighted in bold. The baseline results are taken from Jo et al. (2022) or obtained by running open-source codes. We provide the results of uniqueness, novelty and variance in Section D.1 of the Appendix.

| | QM9 ($|V| \leq 9$) | | | | ZINC250k ($|V| \leq 38$) | | | |
|---|---|---|---|---|---|---|---|---|
| Method | Valid (%)↑ | FCD↓ | NSPDK↓ | Scaf.↑ | Valid (%)↑ | FCD↓ | NSPDK↓ | Scaf.↑ |
| Training set | 100.0 | 0.0398 | 0.0001 | 0.9719 | 100.0 | 0.0615 | 0.0001 | 0.8395 |
| MoFlow (Zang & Wang, 2020) | 91.36 | 4.467 | 0.0169 | 0.1447 | 63.11 | 20.931 | 0.0455 | 0.0133 |
| GraphAF (Shi et al., 2020) | 74.43 | 5.625 | 0.0207 | 0.3046 | 68.47 | 16.023 | 0.0442 | 0.0672 |
| GraphDF (Luo et al., 2021) | 93.88 | 10.928 | 0.0636 | 0.0978 | 90.61 | 33.546 | 0.1770 | 0.0000 |
| EDP-GNN (Niu et al., 2020) | 47.52 | 2.680 | 0.0046 | 0.3270 | 82.97 | 16.737 | 0.0485 | 0.0000 |
| GDSS (Jo et al., 2022) | 95.72 | 2.900 | 0.0033 | 0.6983 | 97.01 | 14.656 | 0.0195 | 0.0467 |
| DiGress (Vignac et al., 2022) | 98.19 | **0.095** | 0.0003 | 0.9353 | 94.99 | 3.482 | 0.0021 | 0.4163 |
| **DruM (Ours)** | **99.69** | 0.108 | **0.0002** | **0.9449** | **98.65** | **2.257** | **0.0015** | **0.5299** |

| | QM9 ($|V| \leq 29$) | | GEOM-DRUGS ($|V| \leq 181$) | |
|---|---|---|---|---|
| Method | Atom Stab.(%) | Mol. Stab.(%) | Atom Stab.(%) | Mol. Stab.(%) |
| G-Schnet (Gebauer et al., 2019) | 95.7 | 68.1 | - | - |
| EN-Flow (Satorras et al., 2021) | 85.0 | 4.9 | 75.0 | 0.0 |
| GDM (Hoogeboom et al., 2022) | 97.0 | 63.2 | 75.0 | 0.0 |
| EDM (Hoogeboom et al., 2022) | 98.7 ±0.1 | 82.0 ±0.4 | 81.3 | 0.0 |
| Bridge (Wu et al., 2022) | 98.7 ±0.1 | 81.8 ±0.2 | 81.0 ±0.7 | 0.0 |
| Bridge+Force (Wu et al., 2022) | 98.8 ±0.1 | 84.6 ±0.2 | 82.4 ±0.7 | 0.0 |
| **DruM (Ours)** | **98.81** ±0.03 | **87.34** ±0.19 | **82.96** ±0.12 | **0.51** ±0.03 |

Figure 3: **(Left) Generation results on the 3D molecule datasets.** Best results are highlighted in bold which is the average of 3 different runs. The baseline results are taken from Hoogeboom et al. (2022) and Wu et al. (2022). **(Right) Convergence of the generative process.** We compare the convergence of the destination mixture from DruM and the implicit destination computed from the predicted noise of EDM. We measure the convergence with L2 distance and further visualize the molecule stability of the predictions through the generative process.

of DruM through the generative process in Figure 2 (Middle), which demonstrates that the local characteristics of the predicted destination rapidly converge to that of the graphs from the training set.

## 4.2 2D MOLECULE GENERATION

We further validate DruM on 2D molecule generation tasks to show that it can accurately generate graphs with both the node features and the topologies of the target graphs.

**Datasets and metrics** We evaluate the quality of generated 2D molecules on two molecule datasets used as benchmarks in Jo et al. (2022): **QM9** (Ramakrishnan et al., 2014) and **ZINC250k** (Irwin et al., 2012). Following the evaluation setting of Jo et al. (2022), we evaluate the models with four metrics: **Validity** is the percentage of the valid molecules among the generated without any posthoc correction. **FCD** (Preuer et al., 2018) measures the distance between the sets of molecules in the chemical space. **NSPDK MMD** (Costa & De Grave, 2010) evaluates the quality of the graph structure compared to the test set. **Scaffold similarity** (Scaf.) evaluates the ability to generate similar substructures. We provide more details in Section C.2 of the Appendix.

**Baselines** We compare to the following molecular graph generative models: **MoFlow** (Zang & Wang, 2020) is a one-shot flow-based model. **GraphAF** (Shi et al., 2020) and **GraphDF** (Luo et al., 2021) are autoregressive flow-based model. **EDP-GNN**, **GDSS**, **ConGress**, and **DiGress** are diffusion models previously explained. We describe further details in Section C.2 of the Appendix.

**Results** Table 2 shows that our method achieves the highest validity on all datasets verifying that DruM can generate valid molecules without correction. Further, DruM outperforms the baselines in FCD and NSPDK metrics demonstrating that the molecules synthesized by DruM are similar to the molecule from the training set in both chemical and graph-structure aspects. Especially, DruM achieves the highest scaffold similarity indicating that it is able to generate similar substructures from that of the training set. We visualize the generated molecules in Section E.1 of the Appendix.

## 4.3 3D MOLECULE GENERATION

To show that DruM is able to generate graphs with both continuous and discrete features, we validate it on 3D molecule generation tasks, which come with discrete atom types and continuous coordinates.

**Datasets and metrics**    We evaluate the generated 3D molecules on two standard molecule datasets used as benchmarks in Hoogeboom et al. (2022): **QM9** (Ramakrishnan et al., 2014) (up to 29 atoms) and **GEOM-DRUGS** (Axelrod & Gomez-Bombarelli, 2022) (up to 181 atoms). Following Hoogeboom et al. (2022), both datasets include hydrogen atoms. For GEOM-DRUGS, we select 30 conformations for each molecule with the lowest energy. We evaluate the quality of the generated molecules with two stability metrics: **Atom stability** is the percentage of the atoms with valid valency. **Molecule stability** is the percentage of the generated molecules that consist of stable atoms. We provide more details in Section C.3 of the Appendix.

**Baselines**    We compare DruM against 3D molecule generative models: **G-Schnet** (Gebauer et al., 2019) is an autoregressive model based on the 3d point sets. **EN-Flow** (Satorras et al., 2021) is a flow-based model. **GDM** and **EDM** (Hoogeboom et al., 2022) are denoising diffusion models. **Bridge** (Wu et al., 2022) is a diffusion model based on the diffusion mixture that learns to approximate the drift and **Bridge+Force** (Wu et al., 2022) adds physical force to the drift. For DruM, we follow the training setting of Hoogeboom et al. (2022) using the same architecture of EGNN (Satorras et al., 2021). We describe further implementation details in Section C.3 of the Appendix.

**Results**    As shown in the table of Figure 3, our method yields the highest atom stability compared to all the baselines on both datasets. Furthermore, DruM achieves higher molecule stability since DruM directly models the topology by learning the destination mixture. Moreover, DruM outperforms Bridge+Force (Wu et al., 2022) even though DruM does not require task-dependent prior force in a simulation-free manner. Notably, DruM achieves non-zero molecule stability in the GEOM-DRUGS dataset consisting of large molecules with up to 181 atoms. We visualize the generated molecules that are stable and the generative process of DruM in Section E of the Appendix, demonstrating that DruM can predict the final molecule at an early stage of the process leading to stable molecules. We observe that DruM generates more number of connected molecules shown in Table 8 of the Appendix.

**Stability analysis**    To further investigate the superior performance of DruM in generating more stable molecules, we conduct an analysis of the convergence and stability of DruM. Figure 3 (Right) shows the convergence of the predicted destination from DruM and the implicit destination from EDM computed from the predicted noise. We observe that for DruM, the predicted destinations converge rapidly to the final destination. After the convergence, the stability of DruM increases as it has sufficient steps to calibrate the details to produce valid molecules, which is visualized in the generative process of Figure 19 of the Appendix. As for EDM, the implicit destinations converge slowly since EDM does not explicitly learn the information of the destination, which leads to lower stability. This analysis shows that learning to predict the final result is significantly superior in capturing the correct topology compared to previous diffusion models.

## 4.4    FURTHER ANALYSIS

We conduct an analysis to investigate the advantages of our framework explained in Section 3.2.

**Exploiting the inductive bias**    To validate that exploiting the inductive bias of the data is critical, we compare DruM against a variant of it without an additional function at the last layer in the model. Figure 2 (Right) shows the complexity of the models $s_\theta$ trained on the Planar dataset, which verifies that the transformation at the last layer significantly reduces the complexity of the model for predicting the destination. Especially, the larger complexity gap at the late stage of the diffusion process suggests that exploiting the inductive bias is crucial for learning the exact destination.

**Comparison with learning the drift**    To verify that learning the destination mixture as in our DruM is superior to learning the drift, we compare with Bridge (Wu et al., 2022) which models the drift of the diffusion mixture process. Table 3 shows that DruM outperforms Bridge especially for the molecule stability, since learning the drift is challenging due to its diverging nature, and further cannot model the topology directly. We further validate that learning the drift performs poorly on general graph generation tasks and fails to generate the correct topology in Section D.2 of the Appendix.

**Early stopping for the generative process**    In Figure 2 (Left) and (Middle), the V.U.N. and the MMD results of DruM in the Planar dataset demonstrate that the estimated destination mixture converges to the exact destination at early sampling steps, accurately capturing both the global topology and local graph characteristics. This allows us to early-stop the diffusion process, which reduces the generation time by up to 20% on this task. The generation results on SBM and Proteins datasets in Section D.2 of the Appendix show a similar tendency.

## 5    CONCLUSION

In this work, we proposed a novel diffusion-based graph generation framework, DruM, that explicitly models the topology of the graphs. Unlike existing graph diffusion models that learn to denoise, our framework directly predicts the destination of the generative process as a weighted mean of data, thereby accurately capturing the topologies of the final graphs that need to be generated. Specifically, DruM constructs the generation process as a mixture of diffusion bridges, which is different from the denoising diffusion process, that drives the generation process toward the predicted destination that converges in an early stage. We extensively validated DruM on diverse graph generation tasks, including 2D/3D molecular generation, on which ours significantly outperforms previous graph generation methods. A promising direction would be the generalization to domains other than graphs where the topology of the data is important, such as proteins and manifolds.

## REPRODUCIBILITY STATEMENT

We use Pytorch (Paszke et al., 2019) to implement our method, which we have included our codes in the supplementary material. We have specified implementation details in Section B and Section C.

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

# Appendix

**Organization** The Appendix is organized as follows: In Section A, we provide the derivations of the results from the main paper. In Section B, we explain the details of our generative framework including the training objectives, the sampling method, and the model architectures. In Section C, we provide experimental details for the generation tasks and further present additional experimental results in Section D. In Section E, we visualize the generated graphs and molecules, with visualized generative processes. Finally, in Section F, we discuss the research directions of our work.

## A DERIVATIONS

### A.1 DIFFUSION BRIDGE PROCESSES

Here we derive the Ornstein-Uhlenbeck (OU) bridge process using Doob's h-transform (Doob & Doob, 1984) and show that the Brownian bridge process is a special case of the OU bridge process. We further discuss a general class of bridge processes and explain the advantage of the OU bridge process.

**Ornstein-Uhlenbeck bridge process** First, we consider the simple case when the reference process is given as a standard OU process without a time-dependent diffusion coefficient:

$$\hat{\mathbb{Q}} \; : \; \mathrm{d}\boldsymbol{G}_t = \alpha \boldsymbol{G}_t \mathrm{d}t + \mathrm{d}\mathbf{W}_t, \tag{10}$$

where $\alpha$ is a constant. Then the Doob's h-transform on $\hat{\mathbb{Q}}$ yields the representation of an endpoint-conditioned process $\hat{\mathbb{Q}}^{\boldsymbol{g}} \coloneqq \hat{\mathbb{Q}}(\cdot | \boldsymbol{G}_T = \boldsymbol{g})$ defined by the following SDE:

$$\hat{\mathbb{Q}}^{\boldsymbol{g}} \; : \; \mathrm{d}\boldsymbol{G}_t = \Big[\alpha \boldsymbol{G}_t + \nabla_{\boldsymbol{G}_t} \log \hat{p}_{T|t}(\boldsymbol{g}|\boldsymbol{G}_t)\Big]\mathrm{d}t + \mathrm{d}\mathbf{W}_t, \tag{11}$$

where $\hat{p}_{T|t}(\boldsymbol{g}|\boldsymbol{G}_t)$ is the transition probability from time $t$ to $T$ of the standard OU process in Eq. (10). Since the standard OU process has a linear drift, the transition probability is Gaussian, i.e. $\hat{p}_{T|t}(\boldsymbol{g}|\boldsymbol{G}_t) = \mathcal{N}(\boldsymbol{g}; \mu_t, \boldsymbol{\Sigma}_t)$, where the mean $\mu_t$ and the covariance $\boldsymbol{\Sigma}_t$ satisfies the following ODEs (derived from the results of Eq.(5.50) and Eq.(5.51) of Särkkä & Solin (2019)):

$$\frac{\mathrm{d}\mu_t}{\mathrm{d}t} = \alpha \mu_t \;\;, \;\; \frac{\mathrm{d}\boldsymbol{\Sigma}_t}{\mathrm{d}t} = \mathbf{I} + 2\alpha \boldsymbol{\Sigma}_t. \tag{12}$$

The ODE with respect to $\boldsymbol{\Sigma}_t$ can be modified as:

$$\frac{\mathrm{d}}{\mathrm{d}t} e^{-2\alpha t} \boldsymbol{\Sigma}_t = e^{-2\alpha t} \mathbf{I}, \tag{13}$$

which give the following closed-form solutions:

$$\mu_t = \hat{u}_t \boldsymbol{G}_t \;\;, \;\; \boldsymbol{\Sigma}_t = \frac{1}{2\alpha}\left(\hat{u}_t^2 - 1\right)\mathbf{I} \;\; \text{for} \;\; \hat{u}_t = e^{\alpha(T-t)}. \tag{14}$$

Therefore, the SDE representation of the standard OU bridge process with fixed endpoint $\boldsymbol{g}$ is given as follows:

$$\hat{\mathbb{Q}}^{\boldsymbol{g}} \; : \; \mathrm{d}\boldsymbol{G}_t = \left[\alpha \boldsymbol{G}_t + \frac{2\alpha}{1 - \hat{u}_t^{-2}}\Big(\frac{\boldsymbol{g}}{\hat{u}_t} - \boldsymbol{G}_t\Big)\right]\mathrm{d}t + \mathrm{d}\mathbf{W}_t. \tag{15}$$

Now we derive the bridge process for the general OU process with a time-dependent diffusion coefficient defined by the following SDE:

$$\mathbb{Q}: \; \mathrm{d}\boldsymbol{G}_t = \alpha \sigma_t^2 \boldsymbol{G}_t \mathrm{d}t + \sigma_t \mathrm{d}\mathbf{W}_t, \tag{16}$$

where $\sigma_t$ is a scalar function. Since the time change (Section 8.5. of Øksendal (2003)) with $\beta_t = \int_0^t \sigma_\tau^2 \mathrm{d}\tau$ of $\hat{\mathbb{Q}}$ in Eq. (10) is equivalent to $\mathbb{Q}$ of Eq. (16), the transition probability $\tilde{p}_{T|t}(\boldsymbol{g}|\boldsymbol{G}_t)$ of the general OU process satisfies the following:

$$\tilde{p}_{T|t}(\boldsymbol{g}|\boldsymbol{G}_t) = \hat{p}_{\beta_T|\beta_t}(\boldsymbol{g}|\boldsymbol{G}_t) \tag{17}$$

Thereby, the OU bridge process conditioned on the endpoint $\boldsymbol{g}$ is defined by the following SDE:

$$\mathbb{Q}^{\boldsymbol{g}} \; : \; \mathrm{d}\boldsymbol{G}_t = \left[ \alpha\sigma_t^2 \boldsymbol{G}_t + \frac{\sigma_t^2}{v_t}\left(\frac{\boldsymbol{g}}{u_t} - \boldsymbol{G}_t\right) \right] \mathrm{d}t + \sigma_t \mathrm{d}\mathbf{W}_t, \tag{18}$$

where the scalar function $u_t$ and $v_t$ are given as:

$$u_t = e^{\alpha(\beta_T - \beta_t)} = \exp\left(\alpha \int_t^T \sigma_\tau^2 \mathrm{d}\tau\right) \; , \;\; v_t = \frac{1}{2\alpha}(1 - u_t^{-2}). \tag{19}$$

Note that the OU bridge process, also known as the constrained OU process, was studied theoretically in previous works (Corlay, 2013; Peluchetti, 2021; Bortoli et al., 2021a). However, we are the first to validate the effectiveness of the OU bridge processes for modeling the generative process through extensive experiments, especially for the generation of graphs in diverse tasks including the generation of general graphs as well as 2D and 3D molecular graphs.

**Brownian bridge process**  We show that the Brownian bridge process is a special case of the OU bridge process. When the constant $\alpha$ of the OU bridge process approaches 0, the scalar function $u_t$ converges to 1 that leads to the convergence of $v_t$ as follows:

$$v_t = \frac{1}{2\alpha}(1 - u_t^{-2}) = \frac{1}{2\alpha}\left(1 - e^{-2\alpha(\beta_T - \beta_t)}\right) \to \beta_T - \beta_t,$$

which is due to the Taylor expansion of the exponential function. Therefore, the OU bridge process for $\alpha \to 0$ is modeled by the following SDE:

$$\mathbb{Q}_{bb}^{\boldsymbol{g}} \; : \; \mathrm{d}\boldsymbol{G}_t = \frac{\sigma_t^2}{\beta_T - \beta_t}(\boldsymbol{g} - \boldsymbol{G}_t)\,\mathrm{d}t + \sigma_t \mathrm{d}\mathbf{W}_t, \tag{20}$$

which is equivalent to the SDE representation of the Brownian bridge process. Compared to the OU bridge process in Eq. (18), the Brownian bridge process has a simpler SDE representation with less flexibility for designing the generative process as the process is solely determined by the noise schedule $\sigma_t$.

Note that the Brownian bridge is an endpoint-conditioned process with respect to a reference Brownian Motion defined by the following SDE:

$$\mathrm{d}\boldsymbol{G}_t = \sigma_t \mathrm{d}\mathbf{W}_t, \tag{21}$$

which is a diffusion process without drift, and also a special case of the OU process that is used for the reference process of the OU bridge process.

**More bridge processes**  Wu et al. (2022) proposes an approach for designing a more general class of diffusion bridges using the Lyapunov function method. Starting from a simple Brownian bridge $\mathbb{Q}_{bb}^{\boldsymbol{g}}$, we can create a new bridge process by adding an extra drift term as follows:

$$\mathbb{Q}_{bb,f}^{\boldsymbol{g}} \; : \mathrm{d}\boldsymbol{G}_t = \left[ \underbrace{\sigma_t f_t(\boldsymbol{G}_t)}_{\text{extra drift}} + \frac{\sigma_t^2}{\beta_T - \beta_t}(\boldsymbol{g} - \boldsymbol{G}_t) \right] \mathrm{d}t + \sigma_t \mathrm{d}\mathbf{W}_t, \tag{22}$$

$$\text{for} \;\; f_t \;\; \text{satisfying} \;\;\; \mathbb{E}_{\boldsymbol{G} \sim \mathbb{Q}_{bb,f}^{\boldsymbol{g}}}[\|f_t(\boldsymbol{G}_t)\|^2] < \infty. \tag{23}$$

$\mathbb{Q}_{bb,f}^{\boldsymbol{g}}$ of Eq. (22) is still a bridge process with endpoint $\boldsymbol{g}$ since the drift of the Brownian bridge (i.e. Eq. (20)) dominates the extra drift term due to the condition of Eq. (23). Moreover, Wu et al. (2022) introduces problem-dependent prior $f$ inspired by physical energy functions.

These general bridge processes could be used for our DruM to construct a mixture process for modeling the generative process, as described in Section 3.1. If the explicit SDE representation for the general bridges is accessible, the mixture process can be represented by leveraging the diffusion mixture representation, and further the Brownian bridge could be replaced with the OU bridge process.

However, in contrast to constructing the generative process as a mixture of the OU bridge processes, using the mixture of the general bridge processes results in difficulty during training; Training a generative model that approximates the mixture of the general bridge processes requires expensive SDE simulation due to the intractable transition probability. We show through extensive experiments that for our approach, the family of OU bridge processes is sufficient to model the complex generation process while the generative model can be trained in a simulation-free manner.

## A.2 DIFFUSION MIXTURE REPRESENTATION

In this section, we provide the formal definition of the diffusion mixture representation (Brigo, 2008; Peluchetti, 2021).

Consider a collection of diffusion processes $\{\mathbb{Q}^\lambda : \lambda \in \Lambda\}$ defined by the SDEs:

$$\mathbb{Q}^\lambda : \mathrm{d}\boldsymbol{Z}_t^\lambda = \eta^\lambda(\boldsymbol{Z}_t, t)\mathrm{d}t + \sigma_t^\lambda \mathrm{d}\mathbf{W}_t^\lambda \quad , \quad \boldsymbol{Z}_0^\lambda \sim p_0^\lambda \tag{24}$$

where $\mathbf{W}_t^\lambda$ are independent standard Wiener processes and $p_0^\lambda$ are the initial distributions. Denote $p_t^\lambda$ as the marginal density of the process $\mathbb{Q}^\lambda$. Further, define the mixture of marginal densities and the mixture of initial distributions with respect to a mixing distribution $\mathcal{L}$ on the collection $\Lambda$ as follows:

$$p_t(z) = \int_\Lambda p_t^\lambda(z)\mathcal{L}(\mathrm{d}\lambda) \quad , \quad p_0(z) = \int_\Lambda p_0^\lambda(z)\mathcal{L}(\mathrm{d}\lambda), \tag{25}$$

Then there exists a diffusion process that induces a marginal density $p_t$, and the diffusion process is modeled by the following SDE:

$$\mathbb{Q}^\mathcal{L} : \mathrm{d}\boldsymbol{Z}_t = \eta(\boldsymbol{Z}_t, t)\mathrm{d}t + \sigma_t \mathrm{d}\mathbf{W}_t \quad , \quad \boldsymbol{Z}_0 \sim p_0, \tag{26}$$

where the drift and diffusion coefficients are given as the weighted mean of the corresponding coefficients of $\mathbb{Q}^\lambda$ as follows:

$$\eta(z, t) = \int_\Lambda \eta^\lambda(z, t)\frac{p_t^\lambda(z)}{p_t(z)}\mathcal{L}(\mathrm{d}\lambda) \quad , \quad \sigma_t^2 = \int_\Lambda (\sigma_t^\lambda)^2 \frac{p_t^\lambda(z)}{p_t(z)}\mathcal{L}(\mathrm{d}\lambda). \tag{27}$$

Below, we provide a proof of this statement.

**proof.**   It is enough to show that $p_t$ defined in Eq. (25) is the solution to the corresponding Fokker-Planck equation of Eq. (26), which is given as follows:

$$\frac{\partial q_t(z)}{\partial t} = -\nabla_z \cdot \left( q_t(z)\eta(z, t) - \frac{1}{2}\sigma_t^2 \nabla_z q_t(z) \right), \tag{28}$$

where $q_t$ denotes the marginal density of Eq. (26). Using the definition of Eq. (25) and the corresponding Fokker-Planck equations with respect to $\mathbb{Q}^\lambda$ for $\lambda \in \Lambda$, we derive the following result:

$$\begin{aligned}
\frac{\partial p_t(z)}{\partial t} &= \frac{\partial}{\partial t}\int_\Lambda p_t^\lambda(z)\mathcal{L}(\mathrm{d}\lambda) = \int_\Lambda \frac{\partial}{\partial t}p_t^\lambda(z)\mathcal{L}(\mathrm{d}\lambda) \\
&= \int_\Lambda \left[ -\nabla_z \cdot \left( \eta^\lambda(z, t)p_t^\lambda(z) - \frac{1}{2}(\sigma_t^\lambda)^2 \nabla_z p_t^\lambda(z) \right) \right]\mathcal{L}(\mathrm{d}\lambda) \\
&= -\nabla_z \cdot \int_\Lambda \left[ \eta^\lambda(z, t)p_t^\lambda(z) - \frac{1}{2}(\sigma_t^\lambda)^2 \nabla_z p_t^\lambda(z) \right]\mathcal{L}(\mathrm{d}\lambda) \\
&= -\nabla_z \cdot \left( p_t(z)\int_\Lambda \eta^\lambda(z, t)\frac{p_t^\lambda(z)}{p_t(z)}\mathcal{L}(\mathrm{d}\lambda) - \frac{1}{2}\nabla_z \left[ p_t(z)\int_\Lambda (\sigma_t^\lambda)^2 \frac{p_t^\lambda(z)}{p_t(z)}\mathcal{L}(\mathrm{d}\lambda) \right] \right) \\
&= -\nabla_z \cdot \left( p_t(z)\eta(z, t) - \frac{1}{2}\sigma_t^2 \nabla_z p_t(z) \right), \tag{29}
\end{aligned}$$

which proves that $p_t$ is the solution to the Fokker-Planck equation of Eq. (28).

## A.3 OU BRIDGE MIXTURE

Now we use the diffusion mixture representation described in Appendix A.2 to derive the OU bridge mixture. Consider a mixture of the collection of OU bridge processes with endpoints in the data distribution, i.e. $\{\mathbb{Q}^{\boldsymbol{g}} : \boldsymbol{g} \sim \Pi^*\}$. We mix this collection of processes with the data distribution $\Pi^*$ as the mixing distribution, which is represented by the following SDE:

$$\begin{aligned}
\mathbb{Q}^{\Pi^*} : \mathrm{d}\boldsymbol{G}_t &= \left[ \int \left( \alpha\sigma_t^2 \boldsymbol{G}_t + \frac{\sigma_t^2}{v_t}\left( \frac{\boldsymbol{g}}{u_t} - \boldsymbol{G}_t \right) \right)\frac{p_t^{\boldsymbol{g}}(\boldsymbol{G}_t)}{p_t(\boldsymbol{G}_t)}\Pi^*(\mathrm{d}\boldsymbol{g}) \right]\mathrm{d}t + \sigma_t \mathrm{d}\mathbf{W}_t \\
&= \left[ \alpha\sigma_t^2 \boldsymbol{G}_t + \frac{\sigma_t^2}{v_t}\left( \frac{1}{u_t}\int \boldsymbol{g}\frac{p_t^{\boldsymbol{g}}(\boldsymbol{G}_t)}{p_t(\boldsymbol{G}_t)}\Pi^*(\mathrm{d}\boldsymbol{g}) - \boldsymbol{G}_t \right) \right]\mathrm{d}t + \sigma_t \mathrm{d}\mathbf{W}_t \\
&= \left[ \alpha\sigma_t^2 \boldsymbol{G}_t + \frac{\sigma_t^2}{v_t}\left( \frac{1}{u_t}\boldsymbol{D}^{\Pi^*}(\boldsymbol{G}_t, t) - \boldsymbol{G}_t \right) \right]\mathrm{d}t + \sigma_t \mathrm{d}\mathbf{W}_t \tag{30}
\end{aligned}$$

where $p_t(z) = \int p_t^{\boldsymbol{g}}(z)\Pi^*(\mathrm{d}\boldsymbol{g})$ is used for the second equality and the definition of the destination mixture (Eq. (5)) is used for the last equality.

### A.4   DESTINATION MIXTURE FOR ATTRIBUTED GRAPHS

We extend the framework of DruM for the generation of *attributed graphs*. To be specific, an attributed graph $\boldsymbol{G}$ with $N$ nodes is defined by the node features $\boldsymbol{X} \in \mathbb{R}^{N \times F}$ and the adjacency matrix $\boldsymbol{A} \in \mathbb{R}^{N \times N}$, where $F$ is the dimension of the node features and the adjacency represents the topology as well as the edge features. Representing the trajectory of the diffusion process as $\boldsymbol{G}_t = (\boldsymbol{X}_t, \boldsymbol{A}_t) \in \mathbb{R}^{N \times F} \times \mathbb{R}^{N \times N}$ for time $t \in [0, T]$, we can derive the OU bridge mixture $\mathbb{Q}^{\Pi^*}$ for attributed graphs as follows:

$$\begin{cases} \mathrm{d}\boldsymbol{X}_t = \left[\alpha_1 \sigma_{1,t}^2 \boldsymbol{X}_t + \frac{\sigma_{1,t}^2}{v_{1,t}}\left(\frac{\boldsymbol{D}_X(\boldsymbol{G}_t,t)}{u_{1,t}} - \boldsymbol{X}_t\right)\right]\mathrm{d}t + \sigma_{1,t}\,\mathrm{d}\mathbf{W}_{1,t} \\ \mathrm{d}\boldsymbol{A}_t = \left[\alpha_2 \sigma_{2,t}^2 \boldsymbol{A}_t + \frac{\sigma_{2,t}^2}{v_{2,t}}\left(\frac{\boldsymbol{D}_A(\boldsymbol{G}_t,t)}{u_{2,t}} - \boldsymbol{A}_t\right)\right]\mathrm{d}t + \sigma_{2,t}\,\mathrm{d}\mathbf{W}_{2,t} \end{cases} \tag{31}$$

where the noise schedules $\sigma_{1,t}$ and $\sigma_{2,t}$ are scalar functions, $\mathbf{W}_{1,t}$ and $\mathbf{W}_{2,t}$ are independent Wiener processes, and the destination mixtures are given as

$$\begin{pmatrix} \boldsymbol{D}_X(\boldsymbol{G}_t,t) \\ \boldsymbol{D}_A(\boldsymbol{G}_t,t) \end{pmatrix} = \int \boldsymbol{g}\frac{p_t^{\boldsymbol{g}}(\boldsymbol{G}_t)}{p_t(\boldsymbol{G}_t)}\Pi^*(\mathrm{d}\boldsymbol{g}). \tag{32}$$

Notably, $\boldsymbol{D}_X(\boldsymbol{G}_t, t)$ and $\boldsymbol{D}_A(\boldsymbol{G}_t, t)$ are the destination mixtures of the node features and the adjacency matrices, respectively for given graph $\boldsymbol{G}_t$. Thereby, our extended framework allows us to directly model the graph topology via learning $\boldsymbol{D}_A(\cdot, t)$ as well as the node features with $\boldsymbol{D}_X(\cdot, t)$. Especially, the generation process of Eq. (31) is applicable to both the continuous and discrete features. We provide the training objective for the attributed graphs in Section B.2.

### A.5   REVERSE-TIME DIFFUSION PROCESS OF THE OU BRIDGE MIXTURE

Here we derive the reverse-time diffusion process of DruM, i.e. the time reversal of the OU bridge mixture. Since the generative process of DruM transports the prior distribution $\Gamma$ to the data distribution $\Pi^*$, the time reversal of DruM transports $\Pi^*$ to $\Gamma$. We show that it has a similar SDE representation as Eq. (30).

We derive the reverse process of the OU bridge mixture by constructing a mixture of the reverse processes of each OU bridge process. To be precise, for the mixture process $\mathbb{Q} := \int \mathbb{Q}^{\boldsymbol{g}}\mathrm{d}\Pi^*$, the reverse process of $\mathbb{Q}$ denoted as $\overline{\mathbb{Q}}$ is equal to the mixture process $\int \overline{\mathbb{Q}}^{\boldsymbol{x}}\mathrm{d}\Gamma$ where $\overline{\mathbb{Q}}^{\boldsymbol{x}}$ is the reverse process of the bridge process $\mathbb{Q}^{\boldsymbol{g}}$ with starting point $\boldsymbol{x}$. For the simplicity of the representation, we first derive the time-reversal of general bridge processes, where the reference process is given as

$$\mathrm{d}\boldsymbol{G}_t^{ref} = \mu(\boldsymbol{G}_t^{ref}, t) + \sigma_t \mathbf{W}_t, \tag{33}$$

with the marginal density denoted as $\tilde{p}_t$. In order to obtain the reverse-time diffusion process, we leverage the reverse-time SDE representation (Anderson, 1982; Song et al., 2021) as follows:

$$\mathrm{d}\overline{\boldsymbol{G}}_t^{ref} = \left[-\mu(\overline{\boldsymbol{G}}_t^{ref}, T{-}t) + \sigma_{T\text{-}t}^2 \nabla_{\overline{\boldsymbol{G}}_t^{ref}} \log \tilde{q}_t(\overline{\boldsymbol{G}}^{ref})\right]\mathrm{d}t + \sigma_{T\text{-}t}\mathrm{d}\mathbf{W}_t, \tag{34}$$

where $\tilde{q}_t = \tilde{p}_{T-t}$ is the marginal density of the process $\{\overline{\boldsymbol{G}}_t^{ref}\}_{t \in [0,T]}$. Then the bridge process of Eq. (34) with fixed end point $\boldsymbol{x} \sim \Gamma$ can be derived by using the Doob's h-transform (Doob & Doob, 1984) as follows:

$$\overline{\mathbb{Q}}^{\boldsymbol{x}} : \mathrm{d}\overline{\boldsymbol{G}}_t = \left[-\mu(\overline{\boldsymbol{G}}_t, T{-}t) + \sigma_{T\text{-}t}^2 \nabla_{\overline{\boldsymbol{G}}_t} \log \tilde{q}_t(\overline{\boldsymbol{G}}_t) + \sigma_{T\text{-}t}^2 \nabla_{\overline{\boldsymbol{G}}_t} \log \tilde{q}_{T|t}(\boldsymbol{x}|\overline{\boldsymbol{G}}_t)\right]\mathrm{d}t + \sigma_{T\text{-}t}\mathrm{d}\mathbf{W}_t, \tag{35}$$

which is a reverse process for the conditioned process of $\boldsymbol{G}^{ref}$ with starting point $\boldsymbol{x}$ and endpoint $\boldsymbol{g} \sim \Pi^*$ fixed. Here using the fact that $\tilde{q}_t = \tilde{p}_{T-t}$, we can see that

$$\tilde{q}_t(y)\tilde{q}_{T|t}(x|y) = \tilde{q}(\overline{\boldsymbol{G}}_T{=}x, \overline{\boldsymbol{G}}_t{=}y) = \frac{\tilde{q}(\overline{\boldsymbol{G}}_T{=}x, \overline{\boldsymbol{G}}_t{=}y)}{\tilde{q}_T(x)}\tilde{q}_T(x) = \tilde{p}_{T-t|0}(y|x)\tilde{q}_T(x), \tag{36}$$

and since $\nabla_{\overline{\boldsymbol{G}}_t} \log \tilde{q}_T(\boldsymbol{x}) = 0$ for fixed $\boldsymbol{x}$, Eq. (35) can be simplified as follows:

$$\overline{\mathbb{Q}}^{\boldsymbol{x}} : \mathrm{d}\overline{\boldsymbol{G}}_t = \left[ -\mu(\overline{\boldsymbol{G}}_t, T{-}t) + \sigma_{T{-}t}^2 \nabla_{\overline{\boldsymbol{G}}_t} \log \tilde{p}_{T{-}t|0}(\overline{\boldsymbol{G}}_t|\boldsymbol{x}) \right] \mathrm{d}t + \sigma_{T{-}t} \mathrm{d}\mathbf{W}_t. \tag{37}$$

Finally, the mixture of the bridge processes $\{\overline{\mathbb{Q}}^{\boldsymbol{x}} : \boldsymbol{x} \sim \Gamma\}$ can be derived using the diffusion mixture representation as follows:

$$\overline{\mathbb{Q}} : \mathrm{d}\overline{\boldsymbol{G}}_t = \left[ -\mu(\overline{\boldsymbol{G}}_t, t) + \sigma_{T{-}t}^2 \int \nabla_{\overline{\boldsymbol{G}}_t} \log \tilde{p}_{T{-}t|0}(\overline{\boldsymbol{G}}_t|\boldsymbol{x}) \frac{q_t^{\boldsymbol{x}}(\overline{\boldsymbol{G}}_t)}{q_t(\overline{\boldsymbol{G}}_t)} \Gamma(\mathrm{d}\boldsymbol{x}) \right] \mathrm{d}t + \sigma_{T{-}t} \mathrm{d}\mathbf{W}_t, \tag{38}$$

where $q_t^{\boldsymbol{x}}$ is the marginal density of $\overline{\mathbb{Q}}^{\boldsymbol{x}}$ and $q_t$ is the marginal density of the mixture process $\overline{\mathbb{Q}}$ defined as $q_t(\cdot) := \int q_t^{\boldsymbol{x}}(\cdot) \Gamma(\mathrm{d}\boldsymbol{x})$.

Using the result of Eq. (38), now we can derive the time reversal of the OU bridge mixture by setting $\mu(z, t) = \alpha \sigma_t^2 z$. Since the transition distributions of the OU process satisfy the following (we provide closed-form mean and covariance of the transition distribution in Eq. (54)):

$$\tilde{p}_{T{-}t|0}(\boldsymbol{z}|\boldsymbol{x}) = \mathcal{N}\left( \boldsymbol{z}; \ \bar{u}_t \boldsymbol{x}, \ \bar{u}_t^2 \bar{v}_t \mathbf{I} \right) \quad \text{for} \quad \bar{u}_t = \exp\left( \alpha \int_0^{T{-}t} \sigma_\tau^2 \mathrm{d}\tau \right), \ \bar{v}_t = \frac{1}{2\alpha}\left( 1 - \bar{u}_t^{-2} \right), \tag{39}$$

the log gradient of the transition distribution can be computed as follows:

$$\nabla_{\boldsymbol{z}} \log \tilde{p}_{T{-}t|0}(\boldsymbol{z}|\boldsymbol{x}) = -\frac{1}{\bar{u}_t^2 \bar{v}_t}\left( \boldsymbol{z} - \bar{u}_t \boldsymbol{x} \right). \tag{40}$$

Thereby, the reverse-time diffusion process of the OU bridge mixture is given by:

$$\overline{\mathbb{Q}} : \mathrm{d}\overline{\boldsymbol{G}}_t = \left[ -\alpha \sigma_{T{-}t}^2 \overline{\boldsymbol{G}}_t + \frac{\sigma_{T{-}t}^2}{\bar{u}_t^2 \bar{v}_t}\left( \bar{u}_t \boldsymbol{D}^\Gamma(\overline{\boldsymbol{G}}_t, t) - \overline{\boldsymbol{G}}_t \right) \right] \mathrm{d}t + \sigma_{T{-}t} \mathrm{d}\mathbf{W}_t, \quad \overline{\boldsymbol{G}}_0 \sim \Pi^*, \tag{41}$$

where $\boldsymbol{D}^\Gamma(\cdot, t)$ is the destination mixture of $\overline{\mathbb{Q}}$ defined as follows:

$$\boldsymbol{D}^\Gamma(\overline{\boldsymbol{G}}_t, t) = \int \boldsymbol{x} \frac{q_t^{\boldsymbol{x}}(\overline{\boldsymbol{G}}_t)}{q_t(\overline{\boldsymbol{G}}_t)} \Gamma(\mathrm{d}\boldsymbol{x}). \tag{42}$$

Since $\overline{\mathbb{Q}}$ describes the diffusion process from the data distribution to the prior distribution, it can be considered a perturbation process. Further, we can observe that the time reversal of the OU bridge mixture is non-linear with respect to $\overline{\boldsymbol{G}}_t$ in general, and completely different from the forward process (i.e. perturbation process) of denoising diffusion models, i.e. the VESDE or VPSDE (Song et al., 2021).

Note that the reverse process of the OU bridge mixture perfectly transports the data distribution $\Pi^*$ to the arbitrary prior distribution $\Gamma$ in the sense that the terminal distribution exactly matches $\Gamma$ for finite terminal time $T$. On the other hand, the forward process of denoising diffusion models, for example, VPSDE (Song et al., 2021), does not perfectly transport the data distribution to the prior distribution. The terminal distribution of the forward process is approximately Gaussian but not exactly a Gaussian distribution for finite $T$, although the mismatch is small for sufficiently large $T$. This is because the forward process requires infinite $T$ in order to decouple the prior distribution $\Gamma$ from the data distribution $\Pi^*$.

In conclusion, the generative process of DruM is different from denoising diffusion models which naturally follows from the fact that the time reversal of the OU bridge mixture is different from the forward processes of denoising diffusion models.

### A.6 DERIVATION OF THE DESTINATION MIXTURE MATCHING OBJECTIVE

We provide the derivation of our destination mixture matching objective, corresponding to Eq. (8) and Eq. (9). First, we leverage the Girsanov theorem Øksendal (2003) to upper bound the KL divergence

between the data distribution $\Pi^*$ and the terminal distribution of $\mathbb{P}^\theta$ denoted as $p_T^\theta$:

$$D_{KL}(\Pi^* \| p_T^\theta) \leq D_{KL}(\mathbb{Q}^{\Pi^*} \| \mathbb{P}^\theta) \tag{43}$$

$$= D_{KL}(\mathbb{Q}_0^{\Pi^*} \| \mathbb{P}_0^\theta) + \mathbb{E}_{\mathbb{Q}^{\Pi^*}}\left[ \log \frac{d\mathbb{Q}^{\Pi^*}}{d\mathbb{P}^\theta} \right] \tag{44}$$

$$= \mathbb{E}_{\boldsymbol{G} \sim \mathbb{Q}^{\Pi^*}}\left[ -\log p_0^\theta(\boldsymbol{G}_0) + \frac{1}{2}\int_0^T \left\| \sigma_t^{-1}\left( \eta_\theta(\boldsymbol{G}_t, t) - \eta(\boldsymbol{G}_t, t) \right) \right\|^2 dt \right] + C \tag{45}$$

$$= \mathbb{E}_{\boldsymbol{G} \sim \mathbb{Q}^{\Pi^*}}\left[ -\log p_0^\theta(\boldsymbol{G}_0) + \frac{1}{2}\int_0^T \gamma_t^2 \left\| \boldsymbol{s}_\theta(\boldsymbol{G}_t, t) - \boldsymbol{D}^{\Pi^*}(\boldsymbol{G}_t, t) \right\|^2 dt \right] + C, \tag{46}$$

where $p_0^\theta$ is a predetermined prior distribution that is easy to sample from, for instance, Gaussian distribution, and $C$ is a constant independent of $\theta$. Note that the first inequality is known as the data processing inequality. The expectation in Eq. (46) corresponds to Eq. (8).

Furthermore, the expectation of Eq. (46) can be written as follows:

$$\mathbb{E}_{\boldsymbol{G} \sim \mathbb{Q}^{\Pi^*}}\left[ \frac{1}{2}\int_0^T \gamma_t^2 \left\| \boldsymbol{s}_\theta(\boldsymbol{G}_t, t) - \boldsymbol{D}^{\Pi^*}(\boldsymbol{G}_t, t) \right\|^2 dt \right]$$

$$= \mathbb{E}_{\boldsymbol{G} \sim \mathbb{Q}^{\Pi^*}}\left[ \frac{1}{2}\int_0^T \gamma_t^2 \left\| \left( \boldsymbol{s}_\theta(\boldsymbol{G}_t, t) - \boldsymbol{G}_T \right) + \left( \boldsymbol{G}_T - \boldsymbol{D}^{\Pi^*}(\boldsymbol{G}_t, t) \right) \right\|^2 dt \right]$$

$$= \mathbb{E}_{\boldsymbol{G} \sim \mathbb{Q}^{\Pi^*}}\left[ \frac{1}{2}\int_0^T \gamma_t^2 \left\| \boldsymbol{s}_\theta(\boldsymbol{G}_t, t) - \boldsymbol{G}_T \right\|^2 dt \right] + \mathcal{E} + \mathcal{E}^T + C_1, \tag{47}$$

where $\mathcal{E}$ and $C_1$ are defined as:

$$\mathcal{E} = \mathbb{E}_{\boldsymbol{G} \sim \mathbb{Q}^{\Pi^*}}\left[ \frac{1}{2}\int_0^T \gamma_t^2 \left( \boldsymbol{s}_\theta(\boldsymbol{G}_t, t) - \boldsymbol{G}_T \right)^T \left( \boldsymbol{G}_T - \boldsymbol{D}^{\Pi^*}(\boldsymbol{G}_t, t) \right) dt \right],$$
$$C_1 = \mathbb{E}_{\boldsymbol{G} \sim \mathbb{Q}^{\Pi^*}}\left[ \frac{1}{2}\int_0^T \gamma_t^2 \left\| \boldsymbol{G}_T - \boldsymbol{D}^{\Pi^*}(\boldsymbol{G}_t, t) \right\|^2 dt \right]. \tag{48}$$

From the definition of the destination mixture (Eq. (5)), the following identity holds for all $t \in [0, T]$:

$$\mathbb{E}_{\boldsymbol{G} \sim \mathbb{Q}^{\Pi^*}} \boldsymbol{D}^{\Pi^*}(\boldsymbol{G}_t, t) = \mathbb{E}_{\boldsymbol{G} \sim \mathbb{Q}^{\Pi^*}} \boldsymbol{G}_T, \tag{49}$$

which gives the following result:

$$\mathcal{E} = \mathbb{E}_{\boldsymbol{G} \sim \mathbb{Q}^{\Pi^*}}\left[ \frac{1}{2}\int_0^T \gamma_t^2 \left( \boldsymbol{s}_\theta(\boldsymbol{G}_t, t) - \boldsymbol{G}_T \right)^T \left( \boldsymbol{G}_T - \boldsymbol{D}^{\Pi^*}(\boldsymbol{G}_t, t) \right) dt \right] \tag{50}$$

$$= \mathbb{E}_{\boldsymbol{G} \sim \mathbb{Q}^{\Pi^*}}\left[ \frac{1}{2}\int_0^T \gamma_t^2 \left( \boldsymbol{s}_\theta(\boldsymbol{G}_t, t) - \boldsymbol{G}_T \right)^T \left( \boldsymbol{G}_T - \boldsymbol{G}_T \right) dt \right] = 0 \tag{51}$$

Therefore, we can conclude that minimizing Eq. (46) is equivalent to minimizing the following loss:

$$\mathbb{E}_{\boldsymbol{G} \sim \mathbb{Q}^{\Pi^*}}\left[ \frac{1}{2}\int_0^T \gamma_t^2 \left\| \boldsymbol{s}_\theta(\boldsymbol{G}_t, t) - \boldsymbol{G}_T \right\|^2 dt \right] \tag{52}$$

which corresponds to the destination mixture matching presented in Eq. (9).

### A.7 ANALYTICAL COMPUTATION OF THE PROBABILITY

In order to practically use the denoising mixture matching (Eq. (9)), we provide the analytical form of the probability $p_{t|0,T}(\boldsymbol{G}_t | \boldsymbol{G}_0, \boldsymbol{G}_T)$. Notice that by construction, the OU bridge mixture with a fixed starting point $\boldsymbol{G}_0$ and an endpoint $\boldsymbol{G}_T$ coincides with the reference OU process in Eq. (16) with a fixed starting point $\boldsymbol{G}_0$ and an endpoint $\boldsymbol{G}_T$. Thereby, $p_{t|0,T}(\boldsymbol{G}_t | \boldsymbol{G}_0, \boldsymbol{G}_T)$ is equal to $\tilde{p}_{t|0,T}(\boldsymbol{G}_t | \boldsymbol{G}_0, \boldsymbol{G}_T)$ where $\tilde{p}$ denotes the marginal probability of the reference OU process of Eq. (16). Using the Bayes theorem, we can derive the following:

$$\tilde{p}(\boldsymbol{G}_t | \boldsymbol{G}_0, \boldsymbol{G}_T) = \frac{\tilde{p}(\boldsymbol{G}_t, \boldsymbol{G}_T | \boldsymbol{G}_0)}{\tilde{p}(\boldsymbol{G}_T | \boldsymbol{G}_0)} = \frac{\tilde{p}(\boldsymbol{G}_T | \boldsymbol{G}_t, \boldsymbol{G}_0)\, \tilde{p}(\boldsymbol{G}_t | \boldsymbol{G}_0)}{\tilde{p}(\boldsymbol{G}_T | \boldsymbol{G}_0)} = \frac{\tilde{p}(\boldsymbol{G}_T | \boldsymbol{G}_t)\, \tilde{p}(\boldsymbol{G}_t | \boldsymbol{G}_0)}{\tilde{p}(\boldsymbol{G}_T | \boldsymbol{G}_0)}, \tag{53}$$

where the last equality is due to the Markov property of the OU process. Note that the transition distributions of the reference OU process are Gaussian with the mean and the covariance as follows:

$$\tilde{p}_{b|a}(\boldsymbol{G}_b|\boldsymbol{G}_a) = \mathcal{N}(\boldsymbol{G}_b; u_{b|a}\boldsymbol{G}_a, u_{b|a}^2 v_{b|a}\mathbf{I}) \text{ for } 0 \le a < b \le T,$$

$$\text{where } u_{b|a} := \exp\left(\alpha \int_a^b \sigma_\tau^2 d\tau\right) , \quad v_{b|a} := \frac{1}{2\alpha}\left(1 - u_{b|a}^{-2}\right). \tag{54}$$

Therefore, the probability $p(\boldsymbol{G}_t|\boldsymbol{G}_0, \boldsymbol{G}_T)$ is also Gaussian resulting from the product of Gaussian distributions, where the mean $\mu_t^*$ and the covariance $\Sigma_t^*$ have analytical forms as follows:

$$\mu_t^* = \frac{v_{T|t}}{u_{t|0}v_{T|0}}\boldsymbol{G}_0 + \frac{v_{t|0}}{u_{T|t}v_{T|0}}\boldsymbol{G}_T , \quad \Sigma_t^* = \frac{v_{T|t}v_{t|0}}{v_{T|0}}\mathbf{I}. \tag{55}$$

The mean and the covariance can be simplified by using the hyperbolic sine function as follows:

$$\mu_t^* = \frac{\sinh{(\varphi_T - \varphi_t)}}{\sinh{(\varphi_T)}}\boldsymbol{G}_0 + \frac{\sinh{(\varphi_t)}}{\sinh{(\varphi_T)}}\boldsymbol{G}_T, \quad \Sigma_t^* = \frac{1}{\alpha}\frac{\sinh{(\varphi_T - \varphi_t)}\sinh{(\varphi_t)}}{\sinh{(\varphi_T)}}\mathbf{I}, \tag{56}$$

where $\varphi_t := \alpha\beta_t = \alpha\int_0^t \sigma_\tau^2 d\tau$.

## A.8 DruM as a stochastic interpolant

Recently, Albergo et al. (2023) introduced the concept of *stochastic interpolant* which unifies the framework for diffusion models from the perspective of continuous-time stochastic processes.

From the results of Eq. (56), we can represent the OU bridge mixture as a stochastic interpolant between the distributions $\Gamma$ and $\Pi^*$ as follows:

$$\boldsymbol{G}_t = \frac{\sinh{(\varphi_T - \varphi_t)}}{\sinh{(\varphi_T)}}\boldsymbol{G}_0 + \frac{\sinh{(\varphi_t)}}{\sinh{(\varphi_T)}}\boldsymbol{G}_T + \left(\frac{1}{\alpha}\frac{\sinh{(\varphi_T - \varphi_t)}\sinh{(\varphi_t)}}{\sinh{(\varphi_T)}}\right)^{1/2}\boldsymbol{Z}. \tag{57}$$

where $\boldsymbol{G}_0$, $\boldsymbol{G}_T$, and $\boldsymbol{Z}$ are random variables sampled independently from the distributions $\Gamma$, $\Pi^*$, and $\mathcal{N}(\boldsymbol{0}, \mathbf{I})$, respectively. Eq. (57) shows that $\boldsymbol{G}_t$ is linear in both the starting point $\boldsymbol{G}_0 \sim \Gamma$ and the endpoint $\boldsymbol{G}_T \sim \Pi^*$. Note that our proposed destination mixture matching is different from the loss introduced in Albergo et al. (2023), as destination mixture matching does not require estimation of the score function. Additionally, we further derive the score function of our DruM in Section A.10.

## A.9 Understanding the informative prior as regularizing the destination mixture

Wu et al. (2022) introduces incorporating prior information into the generative process, for example injecting physical and statistical information. To be specific, given a generative process:

$$d\boldsymbol{G}_t = \eta(\boldsymbol{G}_t, t)dt + \sigma_t\mathbf{W}_t,$$

Wu et al. (2022) modifies the drift by adding a prior function $f(\cdot, t)$ as follows:

$$d\boldsymbol{G}_t = \Big(\underbrace{\sigma_t f(\boldsymbol{G}_t, t) + \eta(\boldsymbol{G}_t, t)}_{\eta_R(\boldsymbol{G}_t, t)}\Big)dt + \sigma_t\mathbf{W}_t, \tag{58}$$

where $f(\cdot, t)$ is designed to be a force defined as $f(\cdot, t) = -\nabla\boldsymbol{E}(\cdot)$ where $E(\cdot)$ is a problem-dependent energy function. Although Wu et al. (2022) shows that incorporating prior information is beneficial for the generation of stable molecules or realistic 3D point clouds, how this modification leads to better performance was not fully explained.

Notably, from the perspective of our framework, we can interpret the incorporation of the prior information as modifying the generative path toward an energy-regularized destination. To be precise, given a generative process modeled by the OU bridge mixture as in Eq. (30), adding the prior function $f(\cdot, t)$ to the drift can be written as follows:

$$\eta_R(\boldsymbol{G}_t, t) = \alpha\sigma_t^2\boldsymbol{G}_t + \frac{\sigma_t^2}{v_t}\left[\frac{1}{u_t}\Big(\boldsymbol{D}^{\Pi^*}(\boldsymbol{G}_t, t) + \frac{u_t v_t}{\sigma_t}f(\boldsymbol{G}_t, t)\Big) - \boldsymbol{G}_t\right], \tag{59}$$

which is equivalent to regularizing the destination mixture with the weighted prior function as follows:

$$\boldsymbol{D}_R^{\Pi^*}(\boldsymbol{G}_t, t) := \boldsymbol{D}^{\Pi^*}(\boldsymbol{G}_t, t) + \frac{u_t v_t}{\sigma_t} f(\boldsymbol{G}_t, t). \tag{60}$$

Since the weight of the prior function converges to 0 through the generative process:

$$\frac{u_t v_t}{\sigma_t} = \frac{\exp\left(\alpha \int_t^T \sigma_\tau^2 \mathrm{d}\tau\right) - \exp\left(-\alpha \int_t^T \sigma_\tau^2 \mathrm{d}\tau\right)}{2\alpha\sigma_t} \to 0 \quad \text{as} \quad t \to T,$$

we can see that $\boldsymbol{D}_R^{\Pi^*}$ converges to the original destination mixture $\boldsymbol{D}^{\Pi^*}$ where the convergence is determined by the prior function. By defining $f(\cdot, t) = -\nabla E(\cdot)$ where $E$ is an energy function, for example, potential energy for the 3D molecules or Riesz energy for the 3D point cloud, the regularized destination mixture has the following representation:

$$\boldsymbol{D}_R^{\Pi^*}(\boldsymbol{G}_t, t) = \boldsymbol{D}^{\Pi^*}(\boldsymbol{G}_t, t) - \frac{u_t v_t}{\sigma_t} \nabla E(\boldsymbol{G}_t). \tag{61}$$

Thereby, $\boldsymbol{D}_R^{\Pi^*}$ follows a path that minimizes the energy function $E$ through the generative process. Therefore, the generative process is guided toward the regularized destination mixture which results in samples that achieve desired physical properties, for instance, stable 3D-structured molecules or point clouds.

## A.10 Associated probability flow ODE of DruM

Since we have derived the reverse-time diffusion process of the OU bridge mixture in Section A.5, we can further derive its associated probability flow ODE (Song et al., 2021), i.e. a deterministic process that shares the same marginal density with the OU bridge mixture.

First, the OU bridge mixture is modeled by the following SDE:

$$\mathrm{d}\boldsymbol{G}_t = \left[\alpha\sigma_t^2 \boldsymbol{G}_t + \frac{\sigma_t^2}{v_t}\left(\frac{1}{u_t}\boldsymbol{D}^{\Pi^*}(\boldsymbol{G}_t, t) - \boldsymbol{G}_t\right)\right]\mathrm{d}t + \sigma_t \mathrm{d}\mathbf{W}_t,$$

where the scalar functions $u_t$ and $v_t$, and the destination mixture $\boldsymbol{D}^{\Pi^*}$ are defined as:

$$u_t = \exp\left(\alpha \int_t^T \sigma_\tau^2 \mathrm{d}\tau\right), \quad v_t = \frac{1}{2\alpha}(1 - u_t^{-2}), \quad \boldsymbol{D}^{\Pi^*}(\boldsymbol{G}_t, t) = \int \boldsymbol{g} \frac{p_t^{\boldsymbol{g}}(\boldsymbol{G}_t)}{p_t(\boldsymbol{G}_t)} \Pi^*(\mathrm{d}\boldsymbol{g}).$$

Then using the results of Section A.5, the reverse-time diffusion process of the OU bridge mixture is modeled by the following SDE:

$$\mathrm{d}\overline{\boldsymbol{G}}_t = \left[-\alpha\sigma_{T-t}^2 \overline{\boldsymbol{G}}_t + \frac{\sigma_{T-t}^2}{\bar{u}_t^2 \bar{v}_t}\left(\bar{u}_t \boldsymbol{D}^\Gamma(\overline{\boldsymbol{G}}_t, t) - \overline{\boldsymbol{G}}_t\right)\right]\mathrm{d}t + \sigma_{T-t}\mathrm{d}\mathbf{W}_t,$$

where the scalar functions $\bar{u}_t$ and $\bar{v}_t$, and the reversed destination mixture $\boldsymbol{D}^\Gamma$ are defined as:

$$\bar{u}_t = \exp\left(\alpha \int_0^{T-t} \sigma_\tau^2 \mathrm{d}\tau\right), \quad \bar{v}_t = \frac{1}{2\alpha}\left(1 - \bar{u}_t^{-2}\right), \quad \boldsymbol{D}^\Gamma(\overline{\boldsymbol{G}}_t, t) = \int \boldsymbol{x} \frac{q_t^{\boldsymbol{x}}(\overline{\boldsymbol{G}}_t)}{q_t(\overline{\boldsymbol{G}}_t)} \Gamma(\mathrm{d}\boldsymbol{x}).$$

From the relation between the diffusion process and its reverse-time diffusion process (for instance, Eq. (33) and Eq. (34)), the score function of the OU bridge mixture can be computed as follows:

$$\nabla_{\boldsymbol{G}_t} \log p_t(\boldsymbol{G}_t) = \frac{1}{v_t}\left(\frac{1}{u_t}\boldsymbol{D}^{\Pi^*}(\boldsymbol{G}_t, t) - \boldsymbol{G}_t\right) + \frac{1}{\bar{u}_{T-t}^2 \bar{v}_{T-t}}\left(\bar{u}_{T-t}\boldsymbol{D}^\Gamma(\boldsymbol{G}_t, T-t) - \boldsymbol{G}_t\right). \tag{62}$$

Therefore, the associated probability flow ODE can be derived as follows:

$$\frac{\mathrm{d}\boldsymbol{G}_t}{\mathrm{d}t} = \alpha\sigma_t^2 \boldsymbol{G}_t + \frac{\sigma_t^2}{v_t}\left(\frac{1}{u_t}\boldsymbol{D}^{\Pi^*}(\boldsymbol{G}_t, t) - \boldsymbol{G}_t\right) - \frac{1}{2}\sigma_t^2 \nabla_{\boldsymbol{G}_t} \log p_t(\boldsymbol{G}_t) \tag{63}$$

$$= \alpha\sigma_t^2 \boldsymbol{G}_t + \frac{\sigma_t^2}{2v_t}\left(\frac{1}{u_t}\boldsymbol{D}^{\Pi^*}(\boldsymbol{G}_t, t) - \boldsymbol{G}_t\right) - \frac{\sigma_t^2}{2\bar{u}_{T-t}^2 \bar{v}_{T-t}}\left(\bar{u}_{T-t}\boldsymbol{D}^\Gamma(\boldsymbol{G}_t, T-t) - \boldsymbol{G}_t\right). \tag{64}$$

To practically use the probability flow ODE as a generative model, the destination mixtures $\boldsymbol{D}^{\Pi^*}(\cdot, t)$ and $\boldsymbol{D}^{\Gamma}(\cdot, t)$ should be approximated by the neural networks $\boldsymbol{s}_\theta(\cdot, t)$ and $\boldsymbol{s}_\phi(\cdot, t)$, respectively. $\boldsymbol{s}_\theta$ can be trained using the destination mixture matching (Eq. (52)). $\boldsymbol{s}_\phi$ also can be trained in a similar way where the trajectories are sampled from the reverse-time process of the OU bridge mixture.

In particular, from the result of Eq. (62), we can see that learning the score function of the mixture process is not interchangeable with learning the destination mixture since the score function additionally requires the knowledge of the reversed destination mixture $\boldsymbol{D}^{\Gamma}$. Our mixture process differs from the denoising diffusion processes for which learning the score function is equivalent to recovering clean data from its corrupted version (Kingma et al., 2021). The difference originates from the difference in the construction of the generative process, where denoising diffusion processes are derived by reversing the forward noising processes while our mixture process is built as a mixture of bridge processes without relying on the time-reversal approach. We further discuss the difference between our framework and the denoising diffusion models in Section A.11.

### A.11    COMPARISON WITH DENOISING DIFFUSION MODELS

Here we explain in detail the difference between our framework based on diffusion mixture and previous denoising diffusion models.

**Comparison of the generative processes**    The main difference with the denoising diffusion models (Ho et al., 2020; Song et al., 2021) is in the different generative processes. While denoising diffusion models derive the generative process by reversing the forward noising process, our DruM constructs the generative process from the mixture of OU bridge processes described in Eq. (2) which does not rely on the time-reversal approach. Due to the difference in the generative process, our DruM demonstrates two distinct properties: First, the mixture process of DruM defines an exact transport from an arbitrary prior distribution to the data distribution by construction. In contrast, denoising diffusion processes are not an exact transport to the data distribution since the forward noising processes require infinitely long diffusion time in order to guarantee convergence to the prior distribution.

Furthermore, our framework does not suffer from the restrictions of denoising diffusion models. Denoising diffusion models require $p_{prior}$ to be approximately independent of the data distribution $\Pi^*$, e.g. Gaussian, as the perturbation process decouples $p_{prior}$ from $\Pi^*$, and further this decoupling requires infinitely long diffusion time $T$. On the contrary, DruM does not have any constraints on the prior distribution $p_{prior}$ and does not require large $T$, since the OU bridge mixture can be defined between two arbitrary distributions for any $T > 0$, where its drift forces the process to the terminal distribution regardless of the initial distribution. Therefore, the OU bridge mixture provides flexibility for our generative framework in choosing the prior distribution and the finite terminal time while maintaining the generative process to be an exact transport from the prior to the data distribution.

**Comparison of the training objectives**    We further compare our training objective in Eq. (52) with the training objectives of denoising diffusion models. First, we clarify that learning the destination mixture is not equivalent to learning the score function for the mixture process of DruM. As derived in Eq. (62) of Section A.10, the score function of the OU bridge mixture additionally requires the knowledge of the reversed destination mixture $\boldsymbol{D}^{\Gamma}$, thus learning the score function needs to predict not only the destination mixture but also the reversed destination mixture. In contrast, the training objectives of denoising diffusion models are interchangeable (Kingma et al., 2021), i.e., learning the score function of the denoising diffusion process is equivalent to recovering clean data from its corrupted version. This difference in the training objective originates from the difference in the generative process, which we have discussed in detail in the previous paragraph.

Furthermore, our training objective differs from the objectives of previous works (Saharia et al., 2022) that aim to recover clean data from its corrupted version. While our DruM learns to predict the destination mixture, i.e. the probable destination represented as the weighted mean of data, Saharia et al. (2022) aims to predict the exact destination which could be problematic as the prediction would be highly inaccurate in early steps which may lead the process in the wrong direction.

It should be noted that the goal of Eq. (52) is to estimate the destination mixture, i.e. the weighted mean of data, not to predict the exact destination as in Saharia et al. (2022). This is because Eq. (52)

Table 3: **Comparison of graph diffusion models.**

| | Explicitly model graph topology | Simulation-free training | Arbitrary prior distribution | Does not require large diffusion time $T$ | Learning object is bounded | Model prediction |
|---|---|---|---|---|---|---|
| EDM (Hoogeboom et al., 2022) | ✗ | ✓ | ✗ | ✗ | ✓ | Noise |
| GDSS (Jo et al., 2022) | ✗ | ✓ | ✗ | ✗ | ✗ | Score |
| Bridge (Wu et al., 2022) | ✗ | ✗ | ✓ | ✓ | ✗ | Drift |
| **DruM** (Ours) | ✓ | ✓ | ✓ | ✓ | ✓ | Data |

is derived from Eq. (8) which minimizes the difference between our model prediction and the ground truth destination mixture. We emphasize that learning the destination mixture of Eq. (8) is only feasible for the OU bridge mixture process in DruM, which cannot be used for denoising diffusion models due to the difference in the generative processes.

In the perspective of the mathematical formulation of the training objective, Eq. (52) differs from the objective of Saharia et al. (2022) in two parts: (1) The computation of the expectation for the squared error loss term is different. The expectation is computed by sampling from the trajectory of the diffusion process, where our DruM uses the OU bridge mixture while previous works use the denoising diffusion process. These two processes are not the same and therefore result in different objectives. (2) The weight function in the loss is different. The weight function $\gamma_t$ of Eq. (8) is different from the weight function used in denoising diffusion models, and $\gamma_t$ is derived to guarantee that minimizing Eq. (52) is equivalent to minimizing the KL divergence between the data distribution and the terminal distribution of our approximated process.

Another line of works on discrete diffusion models (Austin et al., 2021; Hoogeboom et al., 2021; Vignac et al., 2022) aims to predict the probabilities of each state of the final data to be generated. Since these works predict the probabilities, they are limited to data with a finite number of states and cannot be applied to data with continuous features. In contrast, our DruM directly predicts the weighted mean of the data (i.e., destination mixture) instead of the probabilities, which can be applied to data with continuous features, for example, 3D molecules as well as the discrete data, which we experimentally validate to be effective. It is worth noting that our DruM is a continuous diffusion model, and thereby our framework can leverage the advanced sampling strategies that reduce the sampling time or improve sample quality (Campbell et al., 2022), whereas the discrete diffusion models are forced to use a simple ancestral sampling strategy.

## A.12 COMPARISON OF GRAPH DIFFUSION MODELS

We summarize the comparison between closely related graph diffusion models in Table 3.

## B DETAILS OF DRUM

In this section, we provide the details of the training and sampling methods of DruM, describe the models used in our experiments, and further discuss the hyperparameters of DruM.

### B.1 OVERVIEW

We provide the pseudo-code of the training and sampling of our generative framework in Algorithm 1 and 2, respectively. We further provide the implementation details of the training and sampling for each generation task in Section C.

### B.2 TRAINING OBJECTIVES

**Random permutation** The general graph datasets, namely Planar and SBM, contain only 200 graphs. Thus to ensure the permutation invariant nature of the graph dataset, we apply random permutation to the graphs of the training set during training. To be specific, for a graph data $G = (X, A)$ in the training set and random permutation matrix $P$, we use the permuted data $(P^T X, P^T A P)$ for training, where $P^T$ denotes the transposed matrix. We empirically found that this leads to more stable training.

---

**Algorithm 1** Training of DruM

---

**Input:** Model $\boldsymbol{s}_\theta$, constant $\epsilon$
**For each epoch:**
1: Sample graph $\boldsymbol{G}$ from the training set
2: $N \leftarrow$ number of nodes of $\boldsymbol{G}$
3: Sample $t \sim [0, T - \epsilon]$ and $\boldsymbol{G}_0 \sim \mathcal{N}(0, \mathbf{I}_N)$
4: Sample $\boldsymbol{G}_t \sim p_{t|0,T}(\boldsymbol{G}_t|\boldsymbol{G}_0, \boldsymbol{G})$  ▷ Section A.7
5: $\gamma_t \leftarrow \sigma_t / u_t v_t$
6: $\mathcal{L}_\theta \leftarrow \gamma_t^2 \|\boldsymbol{s}_\theta(\boldsymbol{G}_t, t) - \boldsymbol{G}\|^2$  ▷ Eq. (52)
7: Update $\theta$ using $\mathcal{L}_\theta$

---

**Algorithm 2** Sampling of DruM

---

**Input:** Trained model $\boldsymbol{s}_\theta$, number of sampling steps $K$, diffusion step size $\mathrm{d}t$

1: Sample number of nodes $N$ from the training set.
2: $\boldsymbol{G}_0 \sim \mathcal{N}(0, \mathbf{I}_N)$  ▷ Start from noise
3: $t \leftarrow 0$
4: **for** $k = 1$ **to** $K$ **do**
5: $\quad \eta \leftarrow \alpha \sigma_t^2 \boldsymbol{G}_t + \frac{\sigma_t^2}{v_t} \left( \frac{1}{u_t} \boldsymbol{s}_\theta(\boldsymbol{G}_t, t) - \boldsymbol{G}_t \right)$
6: $\quad \mathbf{w} \sim \mathcal{N}(0, \mathbf{I}_N)$
7: $\quad \boldsymbol{G}_{t+\mathrm{d}t} \leftarrow \eta \mathrm{d}t + \sigma_t \sqrt{\mathrm{d}t} \mathbf{w}$  ▷ Euler-Maruyama
8: $\quad t \leftarrow t + \mathrm{d}t$
9: **end for**
10: $\boldsymbol{G} \leftarrow \text{quantize}(\boldsymbol{G}_t)$  ▷ Quantize if necessary
11: **Return:** Graph $\boldsymbol{G}$

---

**Algorithm 3** PC Sampler for DruM

---

**Input:** Trained models $\boldsymbol{s}_\theta$ and $\boldsymbol{s}_\phi$ (described in Section A.10), number of sampling steps $K$, number of correction steps per prediction $M$, diffusion step size $\mathrm{d}t$, score-to-noise ratio $r$
**Output:** Sampled graph $\boldsymbol{G}$
1: Sample number of nodes $N$ from the training set.
2: $\boldsymbol{G}_0 \sim \mathcal{N}(0, \mathbf{I}_N)$  ▷ Start from noise
3: $t \leftarrow 0$
4: **for** $k = 1$ **to** $K$ **do**
5: $\quad \eta \leftarrow \alpha \sigma_t^2 \boldsymbol{G}_t + \frac{\sigma_t^2}{v_t} \left( \frac{1}{u_t} \boldsymbol{s}_\theta(\boldsymbol{G}_t, t) - \boldsymbol{G}_t \right)$
6: $\quad \mathbf{w} \sim \mathcal{N}(0, \mathbf{I}_N)$
7: $\quad \tilde{\boldsymbol{G}}_t \leftarrow \eta \mathrm{d}t + \sigma_t \sqrt{\mathrm{d}t} \mathbf{w}$  ▷ Predictor
8: $\quad$ **for** $m = 1$ **to** $M$ **do**  ▷ Corrector loop
9: $\quad\quad \boldsymbol{D}, \bar{\boldsymbol{D}} \leftarrow \boldsymbol{s}_\theta(\tilde{\boldsymbol{G}}_t, t), \ \boldsymbol{s}_\phi(\tilde{\boldsymbol{G}}_t, T - t)$
10: $\quad\quad s \leftarrow \text{Compute\_Score}(\boldsymbol{D}, \bar{\boldsymbol{D}}, \tilde{\boldsymbol{G}}_t)$  ▷ Eq.(62)
11: $\quad\quad \mathbf{w} \sim \mathcal{N}(0, \mathbf{I}_N)$
12: $\quad\quad \epsilon \leftarrow 2 \left( r \|\mathbf{w}\|_2 / \|s\|_2 \right)^2$
13: $\quad\quad \tilde{\boldsymbol{G}}_t \leftarrow \text{Corrector}(\tilde{\boldsymbol{G}}_t, s, \epsilon)$
14: $\quad$ **end for**
15: $\quad \boldsymbol{G}_{t+\mathrm{d}t} \leftarrow \tilde{\boldsymbol{G}}_t$
16: $\quad t \leftarrow t + \mathrm{d}t$
17: **end for**
18: $\boldsymbol{G} \leftarrow \text{quantize}(\boldsymbol{G}_t)$  ▷ Quantize if necessary
19: **Return:** Graph $\boldsymbol{G}$

---

**Simplified loss**    We provide the explicit form of simplified loss explained in Section 3.2, which uses constant loss coefficient $c$ instead of the time-dependent $\gamma_t$ as follows:

$$\mathcal{L}(\theta) = \mathbb{E}_{\boldsymbol{G} \sim \mathbb{Q}^{\Pi^*}} \left[ \frac{1}{2} \int_0^T c^2 \|\boldsymbol{s}_\theta(\boldsymbol{G}_t, t) - \boldsymbol{G}_T\|^2 \mathrm{d}t \right]. \tag{65}$$

We empirically found that using this loss is beneficial for the generation of continuous features such as eigenvectors of the graph Laplacian or 3D coordinates.

**Attributed graphs**    Especially for the generation of attributed graphs $\boldsymbol{G} = (\boldsymbol{X}, \boldsymbol{A})$, the destination mixture matching for $\boldsymbol{X}$ and $\boldsymbol{A}$ can be derived from our extended framework in Eq. (31). Specifically, for the model $\boldsymbol{s}_\theta(\cdot, t) = (\boldsymbol{s}_\theta^X(\cdot, t), \boldsymbol{s}_\theta^A(\cdot, t))$, we use the following objective:

$$\mathcal{L}(\theta) = \mathbb{E}_{\boldsymbol{G} \sim \mathbb{Q}^{\Pi^*}} \left[ \frac{1}{2} \int_0^T \gamma_{1,t}^2 \left\| \boldsymbol{s}_\theta^X(\boldsymbol{G}_t, t) - \boldsymbol{X}_T \right\|^2 \mathrm{d}t + \frac{\lambda}{2} \int_0^T \gamma_{2,t}^2 \left\| \boldsymbol{s}_\theta^A(\boldsymbol{G}_t, t) - \boldsymbol{A}_T \right\|^2 \mathrm{d}t \right] \tag{66}$$

where $\lambda$ is the hyperparameter. We use $\lambda = 5$ for all our experiments and empirically observed that changing $\lambda$ did not make much difference for sufficient training epochs.

### B.3    MODEL ARCHITECTURE

For the general graph and 2D molecule generation tasks, we leverage the graph transformer network introduced in Dwivedi & Bresson (2020) and Vignac et al. (2022). The node features and the adjacency matrices are updated with multiple attention layers with global features obtained by the self-attention-based FiLM layers (Perez et al., 2018). We additionally use the higher-order adjacency matrices following Jo et al. (2022). For general graph generation tasks, we add the sigmoid function to the output of the model since the entries of the weighted mean of the data are supported in the interval $[0, 1]$. For 2D molecule generation tasks, we apply the softmax function to the output of the node features to model the one-hot encoded atom types, and further apply the sigmoid function to the output of the adjacency matrices. Note that we do not use the structural augmentation as in Vignac et al. (2022). For the 3D molecule generation task, we use EGNN (Satorras et al., 2021) to model the E(3) equivariance of the molecule data. We additionally add a softmax function at the last layer to model the one-hot encoded atom types.

### B.4 Sampling from DruM

Sampling from the generative model requires solving the SDE of Eq. (7). If our model $s_\theta$ can closely approximate the destination mixture, a simple Euler-Maruyama method would be enough to simulate the generative model, which is true for most of the experiments. Since $s_\theta$ is trained on the marginal density $p_t$, $G_t$ outside of $p_t$ could cause incorrect predictions that lead to an undesired destination. To address the limitation, we may leverage the predictor-corrector (PC) sampling method introduced in Song et al. (2021). Using the corrector method such as Langevin dynamics (Song et al., 2021), we force $G_t$ to be drawn from $p_t$ which ensures a more accurate estimation of the destination mixture. The score function to be used for the corrector can be computed as in Eq. (62) of Section A.10. We provide the pseudo-code of the predictor-only sampler and the PC sampler for our DruM in Algorithm 2 and 3. Note that our DruM does not require additional time for sampling compared to the denoising diffusion models, since the generation is equivalent to solving the SDE where the drift is computed each step from the forward pass of the model.

### B.5 Hyperparameters of DruM

The generative process of DruM modeled as the OU bridge mixture is uniquely determined with the noise schedule $\sigma_t$ and constant $\alpha$. Through our experiments, we use $\alpha = -1/2$ and a decreasing linear noise schedule, starting from $\sigma_0^2$ and ends in $\sigma_0^2$ defined as follows:

$$\sigma_t^2 = (1-t)\sigma_0^2 + t\sigma_1^2 \ \text{ for } \ 0 < \sigma_1 < \sigma_0 < 1 \tag{67}$$

We perform a grid search for the hyperparameters $\sigma_0$ and $\sigma_1$ in $\{0.4, 0.6, 0.8, 1.0\}$ and $\{0.1, 0.2, 0.3\}$, respectively, where the search space slightly differs for each generation task.

## C Experimental Details

### C.1 General graph generation

**Datasets and evaluation metrics** We evaluate the quality of generated graphs on three graph datasets used as benchmarks in Martinkus et al. (2022): **Planar** graph dataset consists of 200 synthetic planar graphs where each graph has 64 nodes. We determine that a graph is a valid Planar graph if it is connected and planar. **Stochastic Block Model** (SBM) graph dataset consists of 200 synthetic stochastic block model graphs with the number of communities uniformly sampled between 2 and 5, where the number of nodes in each community is uniformly sampled between 20 and 40. The probability of the inter-community edges and the intra-community edges are 0.3 and 0.05, respectively. We determine that a graph is a valid SBM graph if it has the number of communities between 2 and 5, the number of nodes in each community between 20 and 40, and further using the statistical test introduced in Martinkus et al. (2022). **Proteins** graph dataset (Dobson & Doig, 2003) consists of 918 real protein graphs with up to 500 nodes in each graph. The protein graph is constructed by considering each amino acid as a node and connecting two nodes if the corresponding amino acids are less than 6 Angstrom. For our experiments, we use the datasets provided by Martinkus et al. (2022).

We follow the evaluation setting of Liao et al. (2019) using total variation (TV) distance for measuring MMD which is considerably fast compared to using the earth mover's distance (EMD) kernel, especially for large graphs. Moreover, we use the V.U.N. metric following Martinkus et al. (2022) that measures the proportion of valid, unique, and novel graphs among the generated graphs, where the validness is defined as satisfying the specific property of the dataset explained above.

**Implementation details** We follow the standard setting of Martinkus et al. (2022) using the same data split and evaluation procedures. We report the baseline results taken from Martinkus et al. (2022) and Vignac et al. (2022), or the results obtained from running the open-source codes using the hyperparameters given by the original work. We could not report the results of EDP-GNN (Niu et al., 2020) and DiGress (Vignac et al., 2022) on the Proteins dataset as they took more than 2 weeks. For the diffusion models including our proposed method, we set the diffusion steps to 1000 for a fair comparison.

For our proposed DruM, we train our model for 30,000 epochs for all datasets using a constant learning rate with AdamW optimizer (Loshchilov & Hutter, 2017) and weight decay $10^{-12}$, applying

the exponential moving average (EMA) to the parameters (Song & Ermon, 2020). We perform the hyperparameter search explained in Section B.5 for the lowest MMD and highest V.U.N. results. We initialize the node features with the eigenvectors of the graph Laplacian of the adjacency matrices, which we further scale with constant. During training (Algorithm 1), we sample the noise for the adjacency matrices to be symmetric with zero diagonals. During generation (Algorithm 2), we generate both the node features and adjacency matrices starting from noise, and we quantize the entries of the resulting adjacency matrices. Empirically, we found that the entries of the resulting sample lie very close to the desired values 0 and 1, for which the L1 distance between the resulting sample and the quantized sample is smaller than $10^{-2}$.

In Figure 2 (Left), we measure the Spec. MMD and V.U.N. of our method and the baselines as follows: For DruM we evaluate the destination mixture predicted by DruM. For DiGress, we evaluate the prediction obtained from the predicted probability of each state. For GDSS and ConGress, we evaluate the implicit destination computed from the estimated score or noise following Eq. (16) of Hoogeboom et al. (2022). The Spec. MMD and the V.U.N. are measured after quantizing the predicted adjacency matrix with thresholding at 0.5.

## C.2 2D MOLECULE GENERATION

**Datasets and evaluation metrics** We evaluate the quality of generated 2D molecules on two molecule datasets used as benchmarks in Jo et al. (2022). **QM9** Ramakrishnan et al. (2014) dataset consists of 133,885 molecules with up to 9 heavy atoms of four types. **ZINC250k** Irwin et al. (2012) dataset consists of 249,455 molecules with up to 38 heavy atoms of 9 types. Molecules in both datasets have 3 edge types, namely single bond, double bond, and triple bond. For our experiments, we follow the standard procedure Shi et al. (2020); Luo et al. (2021); Jo et al. (2022) of kekulizing the molecules using the RDKit library (Landrum et al., 2016) and removing the hydrogen atoms from the molecules in the QM9 and ZINC250k datasets.

We evaluate the models with four metrics: **Validity** is the percentage of the valid molecules among the generated without any post-hoc correction such as valency correction or edge resampling. **Fréchet ChemNet Distance** (FCD) (Preuer et al., 2018) measures the distance between the feature vectors of generated molecules and the test set using ChemNet, evaluating the chemical properties of the molecules. **Neighborhood subgraph pairwise distance kernel** (NSPDK) MMD (Costa & De Grave, 2010) measures the MMD between the generated molecular graphs and the molecular graphs from the test set, assessing the quality of the graph structure. **Scaffold similarity** (Scaf.) measures the cosine similarity of the frequencies of Bemis-Murcko scaffolds (Bemis & Murcko, 1996), evaluating the ability to generate similar substructures. See Section C.2 for more details. Among these, FCD and NSPDK MMD metrics measure the distribution similarity between the test dataset and generated samples while scaffold similarity evaluates the similarity of the generated scaffolds.

**Implementation details** We report the results of the baselines taken from Jo et al. (2022), except for the results of the Scaffold similarity (Scaf.) and the results of DiGress, which we obtained by running the open-source codes. Especially, the 2D molecule generation results of DiGress on the QM9 dataset are different compared to the results reported in its original paper, since we have used the preprocessed dataset following the setting of Jo et al. (2022) for a fair comparison with other baselines. We set the diffusion steps to 1000 for all the diffusion models.

For our DruM, we train our model $s_\theta$ with a constant learning rate with AdamW optimizer and weight decay $10^{-12}$, applying the exponential moving average (EMA) to the parameters. We perform the hyperparameter search similar to that of the general graph generation tasks. Especially for DruM, we follow Jo et al. (2022) by using the adjacency matrices in the form of $A \in \{0, 1, 2, 3\}^{N \times N}$ where $N$ is the maximum number of atoms in a molecule and each entries indicating the bond types: 0 for no bond, 1 for the single bond, 2 for the double bond and 3 for the triple bond. Further, we scale the entries with a constant scale of 3 in order to bound the input of the model in the interval $[0, 1]$, and rescale the final sample of the generation process to recover the bond types. We also sample the noise for the adjacency matrices to be symmetric with zero diagonals during training. We quantize the entries of the resulting adjacency matrices to obtain the discrete bond types $\{0, 1, 2, 3\}$. Empirically, we found that the entries of the resulting sample lie very close to the desired bond types $\{0, 1, 2, 3\}$, for which the L1 distance between the resulting sample and the quantized sample is approximately 0.

Table 4: **2D molecule generation results on the QM9 dataset.** The baseline results are taken from Jo et al. (2022) or obtained by running the open-source codes. Best results are highlighted in bold.

| Method | Valid (%)↑ | FCD ↓ | NSPDK ↓ | Scaf. ↑ | Uniq (%) ↑ | Novelty (%) ↑ |
|---|---|---|---|---|---|---|
| MoFlow (Zang & Wang, 2020) | 91.36 ±1.23 | 4.467 ±0.595 | 0.017 ±0.003 | 0.1447 ±0.0521 | 98.65 ±0.57 | **94.72** ±0.77 |
| GraphAF'(Shi et al., 2020) | 74.43 ±2.55 | 5.625 ±0.259 | 0.021 ±0.003 | 0.3046 ±0.0556 | 88.64 ±2.37 | 86.59 ±1.95 |
| GraphDF (Luo et al., 2021) | 93.88 ±4.76 | 10.928 ±0.038 | 0.064 ±0.000 | 0.0978 ±0.1058 | 98.58 ±0.25 | 98.54 ±0.48 |
| EDP-GNN (Niu et al., 2020) | 47.52 ±3.60 | 2.680 ±0.221 | 0.005 ±0.001 | 0.3270 ±0.1151 | **99.25** ±0.05 | 86.58 ±1.85 |
| GDSS (Jo et al., 2022) | 95.72 ±1.94 | 2.900 ±0.282 | 0.003 ±0.000 | 0.6983 ±0.0197 | 98.46 ±0.61 | 86.27 ±2.29 |
| DiGress (Vignac et al., 2022) | 98.19 ±0.23 | **0.095** ±0.008 | 0.0003 ±0.000 | 0.9353 ±0.0025 | 96.67 ±0.24 | 25.58 ±2.36 |
| DruM (ours) | **99.69** ±0.19 | 0.108 ±0.006 | **0.0002** ±0.000 | **0.9449** ±0.0054 | 96.90 ±0.15 | 24.15 ±0.80 |

Table 5: **2D molecule generation results on the ZINC250k dataset.** The baseline results are taken from Jo et al. (2022) or obtained by running the open-source codes. Best results are highlighted in bold.

| Method | Valid (%)↑ | FCD ↓ | NSPDK ↓ | Scaf. ↑ | Uniq (%) ↑ | Novelty (%) ↑ |
|---|---|---|---|---|---|---|
| MoFlow (Zang & Wang, 2020) | 63.11 ±5.17 | 20.931 ±0.184 | 0.046 ±0.002 | 0.0133 ±0.0052 | 99.99 ±0.01 | **100.00** ±0.00 |
| GraphAF (Shi et al., 2020) | 68.47 ±0.99 | 16.023 ±0.451 | 0.044 ±0.005 | 0.0672 ±0.0156 | 98.64 ±0.69 | 99.99 ±0.01 |
| GraphDF (Luo et al., 2021) | 90.61 ±4.30 | 33.546 ±0.150 | 0.177 ±0.001 | 0.0000 ±0.0000 | 99.63 ±0.01 | **100.00** ±0.00 |
| EDP-GNN Niu et al. (2020) | 82.97 ±2.73 | 16.737 ±1.300 | 0.049 ±0.006 | 0.0000 ±0.0000 | 99.79 ±0.08 | **100.00** ±0.00 |
| GDSS (Jo et al., 2022) | 97.01 ±0.77 | 14.656 ±0.680 | 0.019 ±0.001 | 0.0467 ±0.0054 | 99.64 ±0.13 | **100.00** ±0.00 |
| DiGress (Vignac et al., 2022) | 94.99 ±2.55 | 3.482 ±0.147 | 0.0021 ±0.0004 | 0.4163 ±0.0533 | 99.97 ±0.01 | 99.99 ±0.01 |
| DruM (ours) | **98.65** ±0.25 | **2.257** ±0.084 | **0.0015** ±0.0003 | **0.5299** ±0.0441 | 99.97 ±0.03 | 99.98 ±0.02 |

## C.3 3D MOLECULE GENERATION

**Datasets and evaluation metrics** We evaluate the quality of generated 3D molecules on two molecule datasets used as benchmarks in Hoogeboom et al. (2022). **QM9** Ramakrishnan et al. (2014) dataset consists of 133,885 molecules with up to 29 atoms of five types including hydrogen atoms. The node features of the QM9 dataset include the one-hot representation of atom type and atom number. **GEOM-DRUGS** Axelrod & Gomez-Bombarelli (2022) dataset consists of 430,000 molecules with up to 181 atoms of fifteen types including hydrogen atoms. GEOM-DRUGS dataset contains different conformations for each molecule. Among many conformations, the 30 lowest energy conformations for each molecule are retained. For the GEOM-DRUGS dataset, we utilize the one-hot representation of atom type without the atom number. To evaluate the generated molecules, we measure the **atom and molecule stability** by predicting the bond type between atoms with the standard bond lengths and then checking the valency.

**Implementation details** We follow the standard setting of Hoogeboom et al. (2022) for a fair comparison: for the QM9 experiment, we use EGNN with 256 hidden features and 9 layers and train the model, and for the GEOM-DRUGS experiment, we use EGNN with 256 hidden features and 4 layers and train the model. We report the results of the baselines taken from Hoogeboom et al. (2022) and Wu et al. (2022). In Figure 3 (Right), we compute the implicit destination using the estimated noise following Eq. (16) of Hoogeboom et al. (2022).

For our DruM, we train our model $s_\theta$ for 1,300 epochs with batch size 256 for the QM9 experiment, and for 13 epochs with batch size 64 for the GEOM-DRUGS experiment. We apply EMA to the parameters of the model with a coefficient of 0.999 and use AdamW optimizer with learning rate $10^{-4}$ and weight decay $10^{-12}$. The 3D coordinates and charge values are scaled as $\times 4$ and $\times 0.1$, respectively, and we use the simplified loss with a constant $c = 100$. We perform the hyperparameter search with smaller values for coordinates in {0.1, 0.2, 0.3} and higher values for node features in {0.6, 0.7, 0.8, 0.9, 1.0}. For the generation, we use the Euler-Maruyama predictor.

## C.4 COMPUTING RESOURCES

For all experiments, we use NVIDIA GeForce RTX 3090 and 2080 Ti and implement the source code with PyTorch Paszke et al. (2019).

# D ADDITIONAL EXPERIMENTAL RESULTS

## D.1 2D MOLECULE GENREATION

We provide the standard deviation results in Table 4 and Table 5. We additionally report the following two metrics: **Novelty** is the proportion of the molecules generated that are valid and not in the training

Figure 4: **(Left) Generation results on the Planar dataset.** Best results are highlighted in bold, where smaller MMD and larger V.U.N. indicates better results. **(Right) Generated graphs by learning the drift.** The visualized graphs are randomly sampled from the set of generated graphs.

|  | Planar | | | | |
| --- | --- | --- | --- | --- | --- |
|  | Deg. | Clus. | Orbit | Spec. | V.U.N. |
| Training set | 0.0002 | 0.0310 | 0.0005 | 0.0052 | 100.0 |
| GraphRNN (You et al., 2018) | 0.0049 | 0.2779 | 1.2543 | 0.0459 | 0.0 |
| SPECTRE (Martinkus et al., 2022) | 0.0005 | 0.0785 | 0.0012 | 0.0112 | 25.0 |
| EDP-GNN (Niu et al., 2020) | 0.0044 | 0.3187 | 1.4986 | 0.0813 | 0.0 |
| GDSS (Jo et al., 2022) | 0.0041 | 0.2676 | 0.1720 | 0.0370 | 0.0 |
| DiGress (Vignac et al., 2022) | **0.0003** | 0.0372 | **0.0009** | 0.0106 | 75 |
| Drift | 0.0008 | 0.0845 | 0.0075 | 0.0126 | 15 |
| **DruM (Ours)** | 0.0005 | **0.0353** | **0.0009** | **0.0062** | **90.0** |

Figure 5: **Additional MMD results of the destination mixture of DruM** through the generative process. The dotted lines indicate the MMD results of the training set.

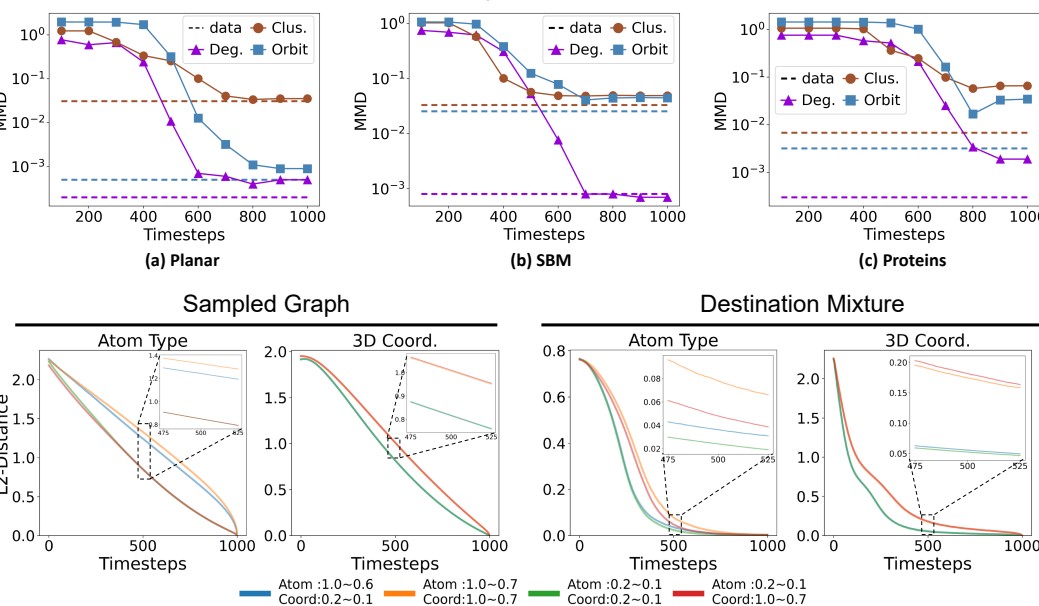

Figure 6: **Convergence of sampled graphs and destination mixtures with varying $\sigma_0$ and $\sigma_1$ values.**

set, and **Uniqueness** is the proportion of the molecules generated that are valid and unique, where valid molecules are the ones that do not violate the chemical valency rule.

## D.2 FURTHER ANALYSIS

**Comparison with learning the drift**    To verify that learning the destination mixture as in our DruM is superior compared to learning the drift, we additionally report the generation result of the variant of DruM which learns the drift, similar to Wu et al. (2022), on the Planar dataset. Table in Figure 4 shows that learning the drift, denoted as Drift in the table, performs poorly generating only 15% valid, novel, and unique graphs. The generated Planar graphs in Figure 4 demonstrate that learning the drift fails to capture the correct topology.

Further, to verify why learning the drift fails to capture the correct topology, we compare the complexity of the models for learning different objectives. As shown in Figure 7, the complexity of learning the drift (Drift) is significantly higher than learning the destination mixture (DruM) for all time steps. Moreover, learning the drift is much harder compared to learning the destination mixture without exploiting the graph structure (w/o Inductive Bias). In particular, the complexity gap dramatically increases at the late stage of the diffusion process, because the drift diverges approaching the terminal time while the destination mixture is supported inside the data space, as discussed in Section 3.2.

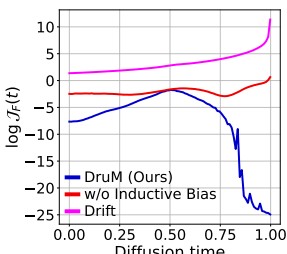

Figure 7: Model complexity comparison of DruM and Drift.

Figure 8: The experimental results for the variant of EDM where it aims to predict the destination (EDM-Dest.). **(Left) Generation results on the 3D molecule QM9 datasets.** Best results are highlighted in bold where the higher stability indicates better results. **(Right) Convergence of the generative process.** We compare the convergence of the destination mixture from DruM, the implicit destination computed from the predicted noise of EDM, and the predicted destination of EDM-Dest. We measure the convergence with L2 distance and further visualize the molecule stability of the predictions through the generative process.

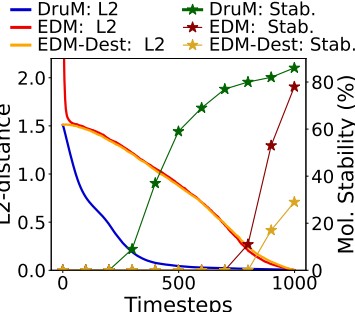

| Method | QM9 $(|V| \leq 29)$ | |
|---|---|---|
| | Atom Stab.(%) | Mol. Stab.(%) |
| G-Schnet (Gebauer et al., 2019) | 95.7 | 68.1 |
| GDM (Hoogeboom et al., 2022) | 97.0 | 63.2 |
| EDM (Hoogeboom et al., 2022) | 98.7 | 82.0 |
| Bridge (Wu et al., 2022) | 98.7 | 81.8 |
| Bridge+Force (Wu et al., 2022) | 98.8 | 84.6 |
| EDM-Dest. | 94.02 | 35.95 |
| **DruM (Ours)** | **98.81** | **87.34** |

**Early stopping for generation process** We provide additional MMD results of the generative processes in Figure 5, which show that the estimated destination mixture converges to the exact destination around 800 diffusion steps for all datasets.

**Role of the diffusion coefficient** We can observe that the generative process of DruM is uniquely determined by the constant $\alpha$ and the diffusion coefficient $\sigma_t$. These two coefficients control the convergence behavior of the diffusion process: large $\alpha$ and small $\sigma_t$ lead to a drift with a large norm that forces the trajectory to converge quickly. Here, we demonstrate the effect of the diffusion coefficient $\sigma_t$ on the convergence of the generative process. Figure 6 (Sampled Graph) shows that the smaller values of $\sigma_t$ (i.e. 0.2~0.1) lead to faster convergence of the trajectory to the final result, compared to the larger $\sigma_t$. This is due to the fast convergence of each bridge process with small $\sigma_t$. Especially, as shown in Figure 6 (Destination Mixture), large $\sigma_t$ for the continuous feature (i.e., 3D coordinates) leads to slower convergence of the destination mixture since it destroys the topology of graphs and makes it hard to predict the final result.

**Destination prediction through EDM** Additionally, we compare our DruM with the variant of EDM Hoogeboom et al. (2022) which learns to predict the final result of the denoising process instead of learning the noise. Table of Figure 8 shows the generation result of this variant, denoted as EDM-Dest., on the 3D molecule QM9 dataset. EDM-Dest. exhibits the lowest atom stability and extremely low molecule stability of less than 40%, which performs significantly worse than DruM as well as the original EDM. This is because EDM-Dest. depends on a single deterministic prediction of the destination during the generative process, and the inaccurate prediction of the final result at the early step of the generative process leads the process in the wrong direction resulting in invalid molecules, as discussed in the Introduction and Section 3.1.

On the other hand, our DruM predicts the destination of the generative process using the destination mixture which represents the probable destination as a weighted mean of the data, thereby guiding the process in the right direction resulting in valid molecules with correct topology. We further provide the convergence results of EDM-Dest. in Figure 8, which demonstrates that the prediction of DruM converges significantly faster than that of EDM and EDM-Dest. The inaccurate prediction of EDM-Dest. results in slower convergence and low molecule stability.

**Analysis on the model architecture** As shown in Table 6 and 7, GDSS using graph transformer architecture shows improved performance over original GDSS but is still outperformed by our DruM with a large margin in V.U.N, FCD, and NSPDK. These results verify that the superior performance of DruM comes from its ability to accurately model the topology of the final graph to be generated.

Table 6: **General graph generation results with GDSS using graph transformer.**

| | Planar | | | | | SBM | | | | |
| | Synthetic, $|V| = 64$ | | | | | Synthetic, $44 \leq |V| \leq 187$ | | | | |
| | Deg. | Clus. | Orbit | Spec. | V.U.N. | Deg. | Clus. | Orbit | Spec. | V.U.N. |
|---|---|---|---|---|---|---|---|---|---|---|
| Training set | 0.0002 | 0.0310 | 0.0005 | 0.0052 | 100.0 | 0.0008 | 0.0332 | 0.0255 | 0.0063 | 100.0 |
| GDSS | 0.0041 | 0.2676 | 0.1720 | 0.0370 | 0.0 | 0.0212 | 0.0646 | 0.0894 | 0.0128 | 5.0 |
| GDSS+Transformer | 0.0036 | 0.1206 | 0.0525 | 0.0137 | 5.0 | 0.0411 | 0.0565 | 0.0706 | 0.0074 | 27.5 |
| ConGress | 0.0048 | 0.2728 | 1.2950 | 0.0418 | 0.0 | 0.0273 | 0.1029 | 0.1148 | - | 0.0 |
| DiGress | **0.0003** | 0.0372 | **0.0009** | 0.0106 | 75 | 0.0013 | 0.0498 | 0.0434 | 0.0400 | 74 |
| **DruM (Ours)** | 0.0005 | **0.0353** | **0.0009** | **0.0062** | **90.0** | **0.0007** | **0.0492** | 0.0448 | **0.0050** | **85.0** |

Table 7: **2D molecule generation results with GDSS using graph transformer.**

| | QM9 $(|V| \leq 9)$ | | | | ZINC250k $(|V| \leq 38)$ | | | |
| Method | Valid (%)↑ | FCD↓ | NSPDK↓ | Scaf.↑ | Valid (%)↑ | FCD↓ | NSPDK↓ | Scaf.↑ |
|---|---|---|---|---|---|---|---|---|
| Training set | 100.0 | 0.0398 | 0.0001 | 0.9719 | 100.0 | 0.0615 | 0.0001 | 0.8395 |
| GDSS | 95.72 | 2.900 | 0.0033 | 0.6983 | 97.01 | 14.656 | 0.0195 | 0.0467 |
| GDSS+Transformer | 99.68 | 0.737 | 0.0024 | 0.9129 | 96.04 | 5.556 | 0.0326 | 0.3205 |
| DiGress | 98.19 | **0.095** | 0.0003 | 0.9353 | 94.99 | 3.482 | 0.0021 | 0.4163 |
| **DruM (Ours)** | **99.69** | 0.108 | **0.0002** | **0.9449** | **98.65** | **2.257** | **0.0015** | **0.5299** |

# E  VISUALIZATION

In this section, we visualize the generated graphs and molecules of DruM, and further provide visualization of the diffusion processes for diverse generation tasks.

## E.1  GENERATED SAMPLES OF DRUM

**General graphs**  Graphs from the training set and the generated graphs of DruM are visualized in Figure 9, 10, and 11. The visualized graphs are randomly selected from the training set and the generated graph set. Note that we visualize the entire graph for the Proteins dataset, unlike Martinkus et al. (2022) which visualizes the largest connected component since it fails to consistently generate connected graphs. For DruM, we found that 92% of the generated Proteins graphs are connected.

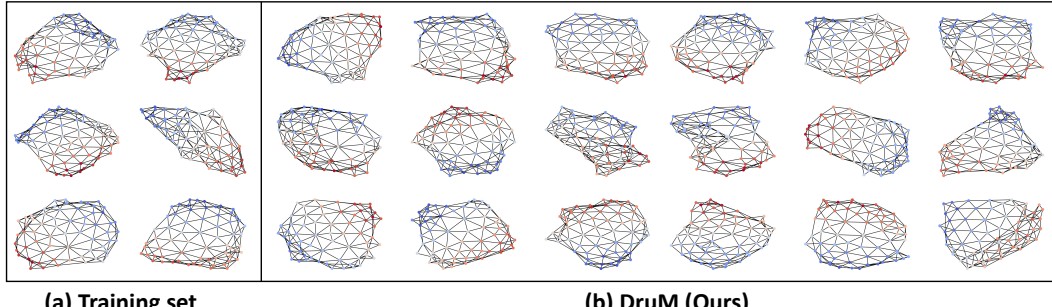

**(a) Training set**  **(b) DruM (Ours)**

Figure 9: **Visualization of graphs from the Planar dataset and the generated graphs of DruM.**

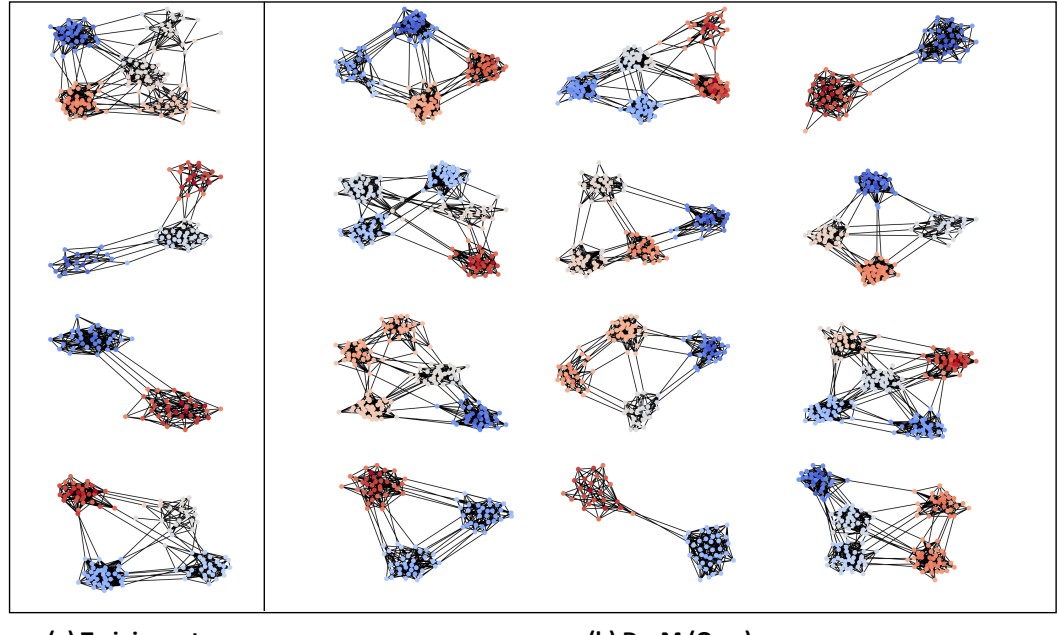

(a) Training set        (b) DruM (Ours)

Figure 10: **Visualization of graphs from the SBM dataset and the generated graphs of DruM.**

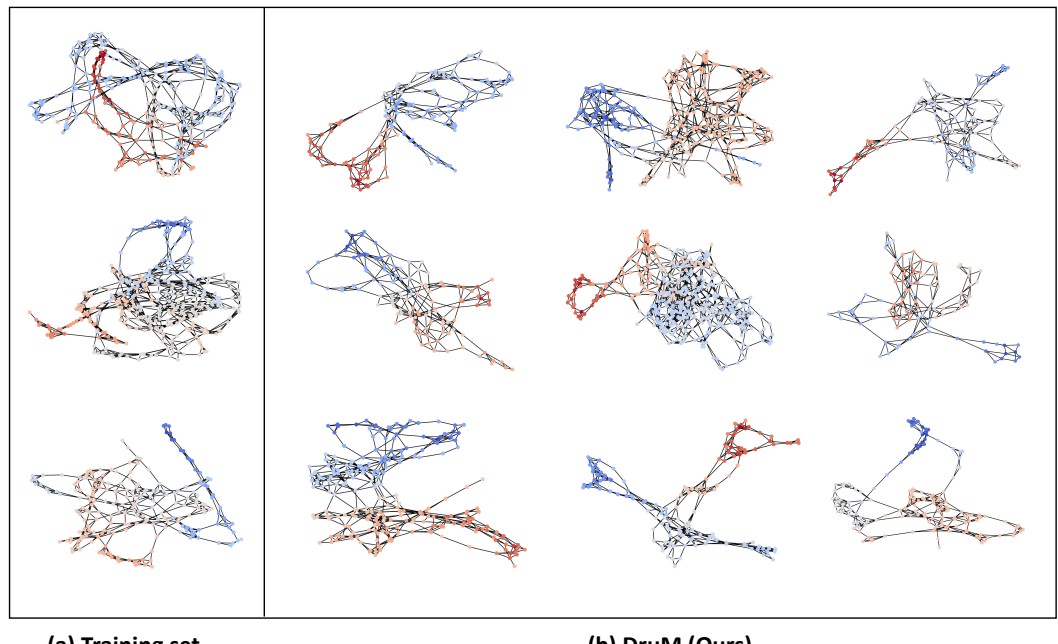

(a) Training set        (b) DruM (Ours)

Figure 11: **Visualization of graphs from the Proteins dataset and the generated graphs of DruM.**

**2D molecules**    We provide the visualization of the molecules from the training set and the generated 2D molecules in Figure 12 and 13. These molecules are randomly selected from the training set and the generated molecule set.

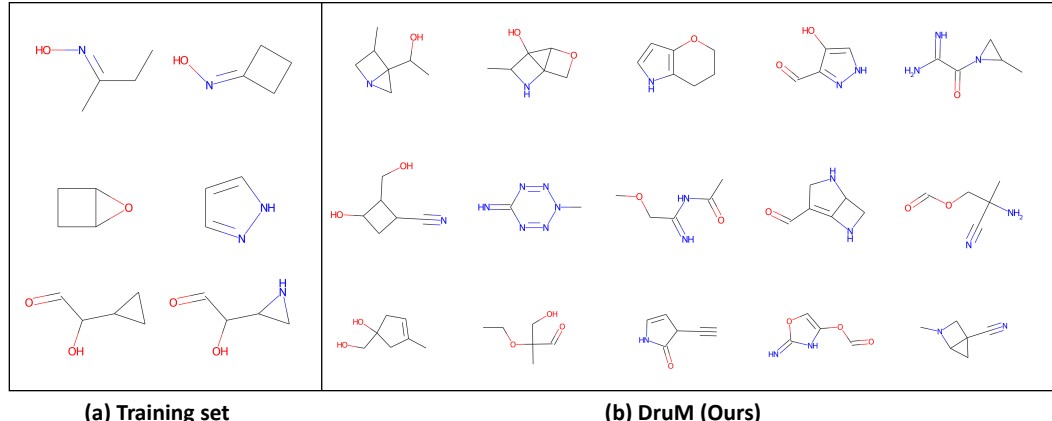

      **(a) Training set**                                **(b) DruM (Ours)**

Figure 12: **Visualization of molecules from the QM9 dataset and the generated molecules of DruM for the 2D molecule generation experiment.**

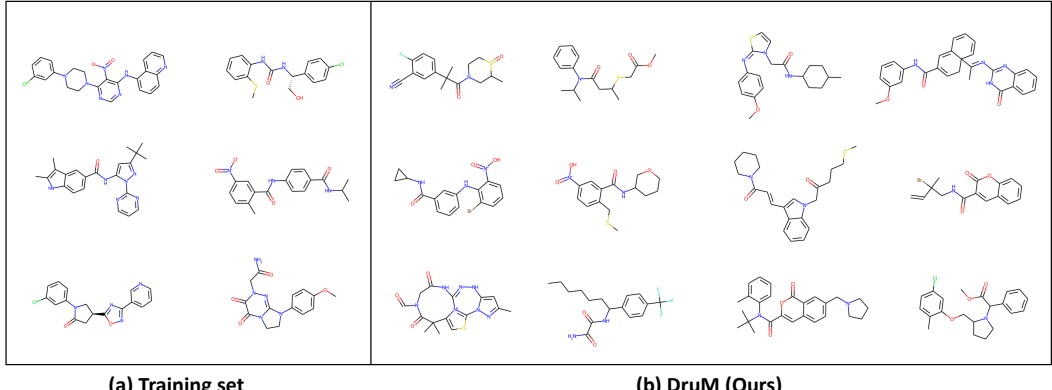

      **(a) Training set**                                **(b) DruM (Ours)**

Figure 13: **Visualization of the molecules from the ZINC250k dataset and the generated molecules of DruM for the 2D molecule generation experiment.**

**3D molecules** We visualize the generated molecules for the 3D molecule generation experiment in Figure 14 and 15. Note that the visualized molecules are all stable. For the GEOM-DRUGS experiment, we observe that a few of the generated molecules are not connected as pointed out in Hoogeboom et al. (2022). To measure how many graphs are connected, we report the fraction of the connected graphs of 3 different runs. Table 8 shows that DruM can generate a significantly larger number of connected molecules compared to EDM Hoogeboom et al. (2022).

Table 8: Fraction of connected graphs on GEOM-DRUGS experiment.

| Methods | Connected (%) |
|---|---|
| EDM | 37.70 ±0.79 |
| **DruM (Ours)** | **56.57** ±0.31 |

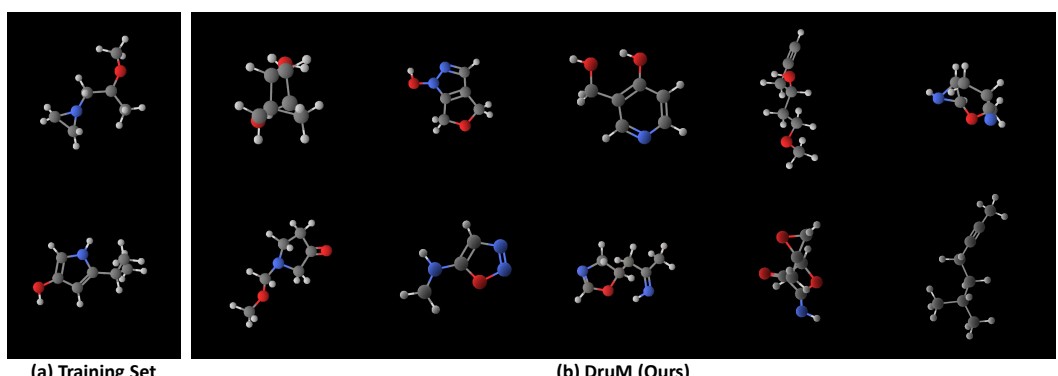

(a) Training Set      (b) DruM (Ours)

Figure 14: **Visualization of the molecules from the QM9 dataset and the generated molecules of DruM for the 3D molecule generation experiment.**

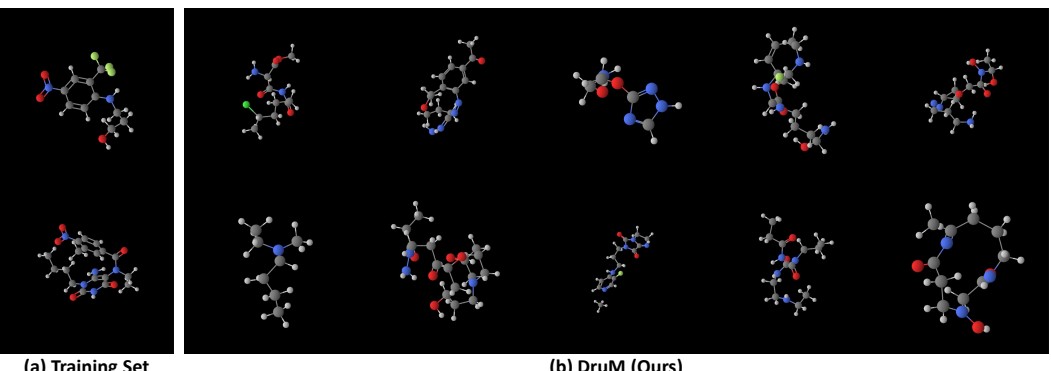

(a) Training Set      (b) DruM (Ours)

Figure 15: **Visualization of the molecules from the GEOM-DRUGS dataset and the generated molecules of DruM for the 3D molecule generation experiment.**

## E.2 GENERATIVE PROCESS OF DRUM

Here we visualize the generative process of DruM. We visualize the generative process of general graphs in Figure 16, 17, and 18. We also visualize the generative process of the 3D molecules in Figure 19.

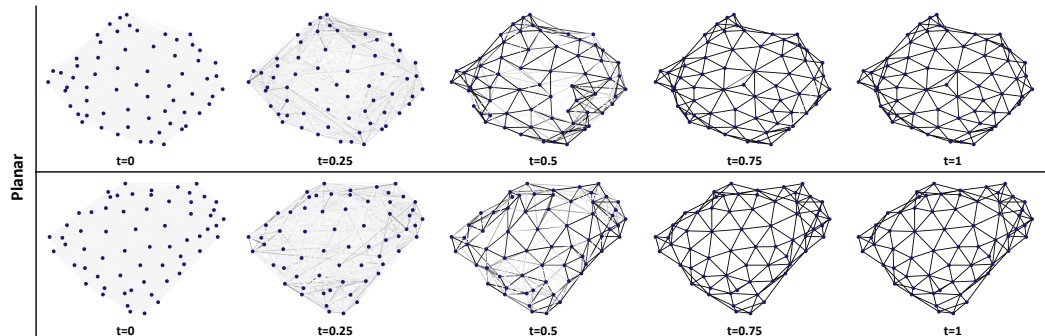

Figure 16: **Visualization of the generative process of DruM.** We visualize the destination mixture from DruM on the Planar dataset.

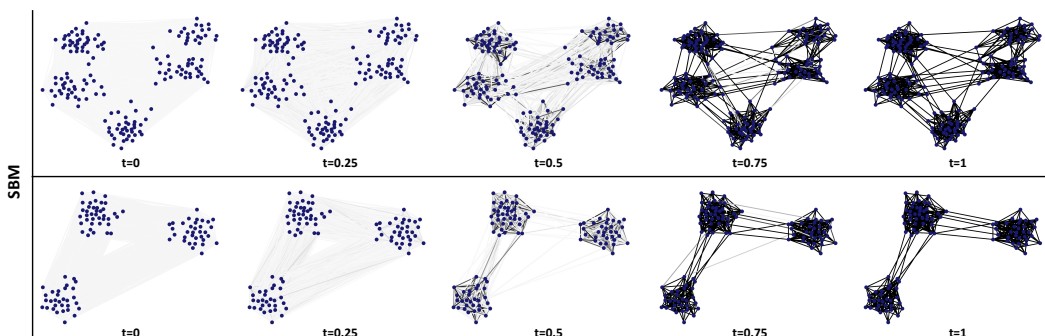

Figure 17: **Visualization of the generative process of DruM.** We visualize the destination mixture from DruM on the SBM dataset.

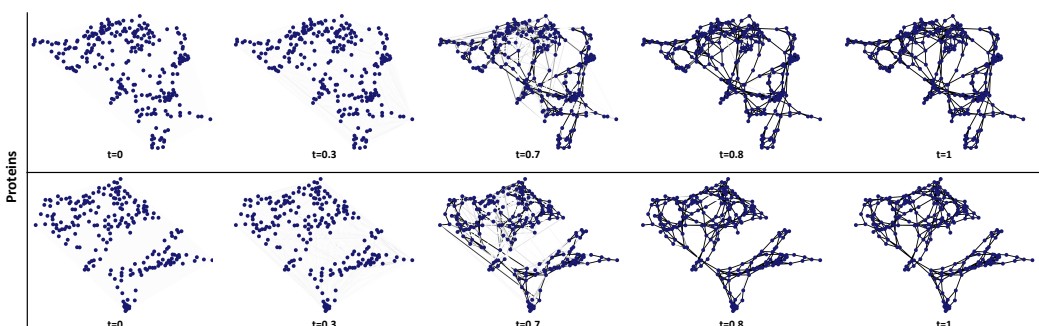

Figure 18: **Visualization of the generative process of DruM.** We visualize the destination mixture from DruM on the Proteins dataset.

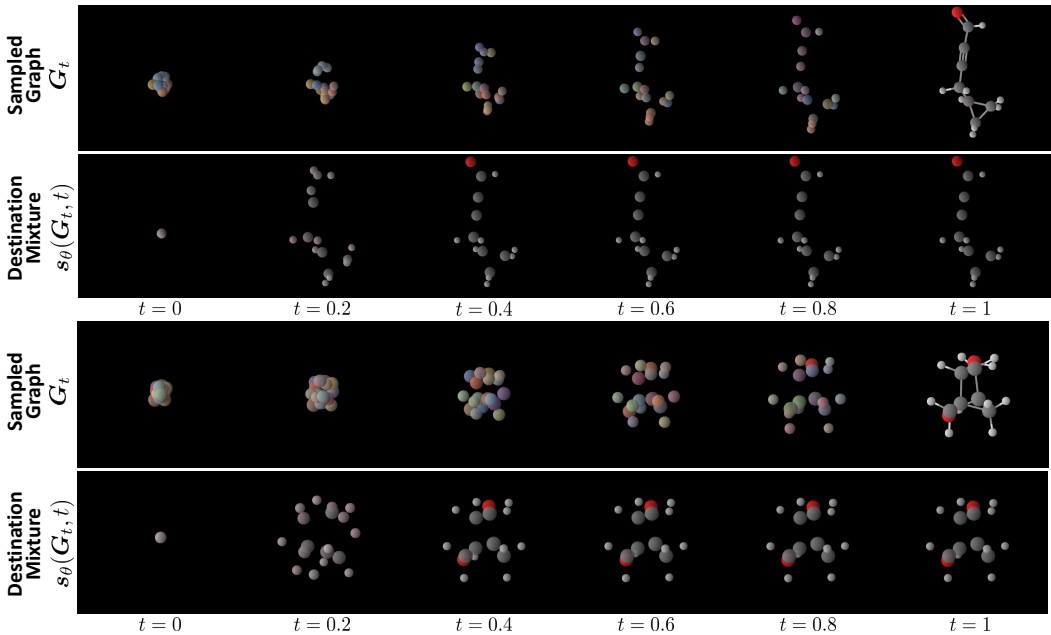

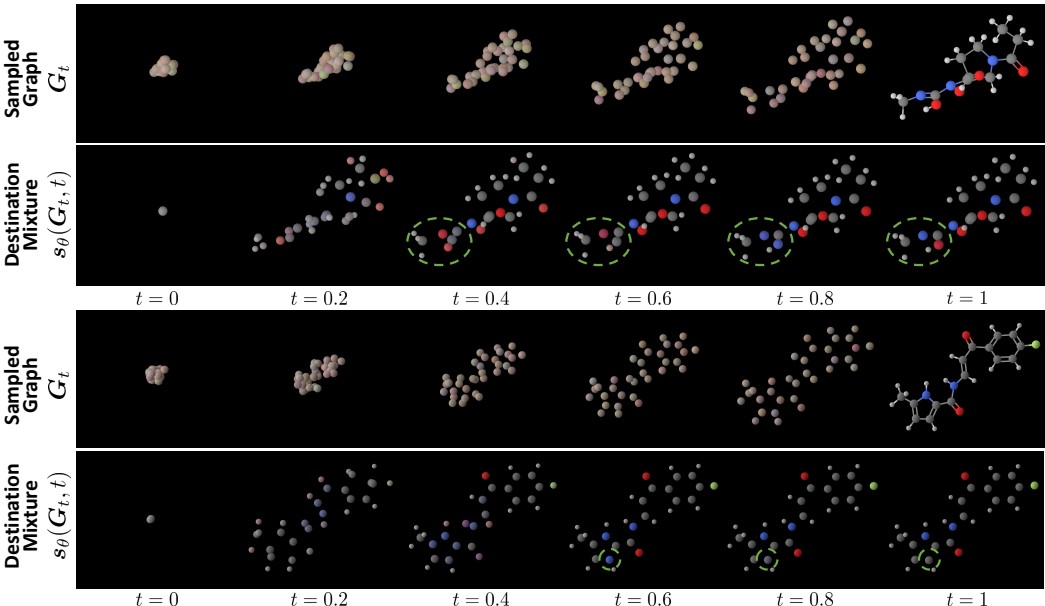

Figure 19: **Visualizations of the 3D molecule generative process** of DruM on QM9 dataset (Top) and GEOM-DRUGS dataset (Bottom). For each dataset, we visualize the trajectory $G_t$ in the first row, and we visualize the estimated destination mixtures from DruM in the second row. Note that the visualized molecules are stable. The atom types and the 3D coordinates of the atoms inside the green circles are calibrated after the convergence of the destination mixtures, where the convergence is achieved at an early stage.

## F  FUTURE WORK

We proposed a novel diffusion-based graph generation framework that directly predicts the destination of the generative process as a weighted mean of data, thereby accurately capturing the topologies of the final graphs that need to be generated. We have shown that our framework is able to generate graphs with correct topology for diverse graph generation tasks, including 2D/3D molecular generation, on which ours significantly outperforms previous graph generation methods. While DruM shows superior performance, future work would benefit from improving our framework.

First, the likelihood of the generative process of DruM cannot be directly computed from the training objective. In order to compute the likelihood, one could derive an associated probability flow ODE of DruM as described in Section A.10, but this requires training an additional model for estimating the reverse destination mixture.

Furthermore, the proposed framework is focused on unconditional graph generation tasks. We could design a conditional framework of DruM by training a model $s_\theta(\boldsymbol{G}_t, t, \boldsymbol{c})$ for a given condition (i.e., class label) $\boldsymbol{c}$ for estimating the $\boldsymbol{c}$-conditional destination mixture defined as follows:

$$\boldsymbol{D}^{\Pi_{\boldsymbol{c}}^*}(\boldsymbol{G}_t, t) := \int \boldsymbol{g} \frac{p_t^{\boldsymbol{g}}(\boldsymbol{G}_t)}{p_t(\boldsymbol{G}_t)} \Pi_{\boldsymbol{c}}^*(\mathrm{d}\boldsymbol{g}), \quad \Pi_{\boldsymbol{c}}^* := \{\boldsymbol{g} : \boldsymbol{g} \sim \Pi^* \text{ with label } \boldsymbol{c}\}. \tag{68}$$

Intuitively, the generative process of the modified OU bridge mixture, for which the destination mixture is replaced by $\boldsymbol{D}^{\Pi_{\boldsymbol{c}}^*}(\boldsymbol{G}_t, t)$ is guided by the conditional destination mixture that terminates in the conditioned distribution $\Pi_{\boldsymbol{c}}^*$. We leave this conditional framework as future work.

