# OpenReview forum: "Graph Generation with Destination-Predicting Diffusion Mixture"
_ICLR.cc/2024/Conference — Submitted to ICLR 2024_

### Official Review · Reviewer_HfR4 · 2023-11-01

**Soundness:** 3 good
**Presentation:** 3 good
**Contribution:** 2 fair
**Rating:** 6
**Confidence:** 3

**Summary:**

This work proposes a novel diffusion-based graph generation framework. The framework leverages the process conditioning on the generation destination, which aims to more accurately capture the graph topology. The training objective in this framework can be approximately represented as to predict a weighted mixture/average of the data.  The empirical evaluation of 2D/3D graph generation shows the effectiveness of the proposed methods.

**Strengths:**

+ This work introduces a new diffusion-based graph generation framework where the generation procedure is conditioned on the destination graph. The framework at a high level combines the benefits of the previous discrete diffusion process for graphs (DiGress) that directly predict the destination, and the continuous diffusion process to handle the potential 3D continuous features that 3D graphs may have.

+ The derivation seems reasonable at a high level. (I have not checked the proof).

+ The empirical results show the effectiveness of the framework.

**Weaknesses:**

- From the technical side, the framework seems to directly leverage the tool from [1][2]. So, the fundamental technique is not novel, though the application to graph generation may be novel. Moreover, the derivation seems to be nothing specified to graph topology but quite generic. So, it is not persuasive that the adopted framework may capture graph topology better than other frameworks. The benefits of practical performance seem to entirely inherit from the previously developed framework [1][2].

- Eq.9 seems to just estimate the exact endpoint $G_T$ instead of the destination mixture $D(G_t, t)$. However, Eq.8 is to estimate $D(G_t, t)$, isn't it? This seems a conflict with the statement just below Eq. 9.

[1] Diffusion-based molecule generation with informative prior bridges, NeurIPS 2022

[2] Learning Diffusion Bridges on Constrained Domains, ICLR 2023

**Questions:**

Please address the two weaknesses: 1) why is the framework related to graph topology? The framework seems to be a direct application of previous frameworks to graph generation; 2) why Eq.9 is to estimate a mixture of data points.

I may re-evaluate this work based on the authors' response.

---

> ### Author Response · Authors · 2023-11-15
> **Initial Response to Reviewer HfR4 (1/2)**
>
> We sincerely thank you for your constructive and helpful comments. We appreciate your positive comments that
> - Our work introduces a new diffusion-based graph generation framework.
> - Our method has benefits that can directly predict the destination of the diffusion process while being able to handle potential continuous features.
> - Experimental results demonstrate the effectiveness of the framework.
>
> We provide an updated revision of the paper highlighted in orange, and we address your concerns in our initial response below.
>
> ---
>
> **Comment 1-1:**
> The framework seems to be a direct application of previous frameworks [1, 2] to graph generation.
>
> [1] Wu et al., Diffusion-based molecule generation with informative prior bridges, NeurIPS 2022.
>
> [2] Liu et al., Learning Diffusion Bridges on Constrained Domains, ICLR 2023.
>
> **Response 1-1:**
> **Our framework is not a direct application of [1, 2]** since **the superior ability of DruM to generate valid graphs comes from our contribution in the parameterization and the training objective**.
>
> In our work, we aim to model the graph topology by directly modeling the graph structure, and thereby our framework learns to **predict the weighted mean of the data**, i.e. destination mixture which is achieved from our **novel parameterization of Eq. (5) and our training objective in Eq. (9)**.
>
> On the other hand, previous works [1, 2] learn to predict the drift function focusing on improving the generative process by adding prior information to the diffusion process [1] or using domain-specific (e.g., bounded, ordinal) bridges [2], which are not related to graphs. Implicitly modeling the topology through estimating drift is sub-optimal for modeling the discreteness of the graph structure as well as the structural properties. As a result, the **direct application of previous works for graph generation fails to generate valid graphs**. We experimentally validate this in Section 4.4:
> - In Figure 4, learning the drift (denoted as Drift) fails to generate planar graphs resulting in low validity compared to our approach.
> - In Figure 3 (Left), previous work [1] (denoted as Bridge) achieves lower molecule stability compared to our DruM on 3D molecule generation.
> - Especially, DruM outperforms Bridge+Force [1] even without the need of adding domain-specific knowledge (i.e., AMBER Inspired Physical Energy), and achieving $\\times$17.5 faster training compared to [1] due to simulation-free training.
>
> Moreover, the difference in the objectives results in a significant difference in training models. **Learning the destination mixture (ours)** is considerably easier compared to learning the drift function ([1,2]) as the drift function diverges approaching the terminal time whereas the destination mixture is supported inside the bounded data space (as explained at the end of Section 3.2). Figure 7 (Appendix D.2) of the updated revision shows that the complexity of the model estimating the destination mixture (ours) is significantly lower than that of estimating the drift function (previous works), especially for the later stage of the diffusion process where the drift function diverges.
>
> Notably, our work provides novel technical contributions that are not covered by the related works as follows:
> - We propose to construct the generative process as a **mixture of the OU bridge process (Eq. (2))**, whereas previous works use the Brownian bridge process which is a special case of our OU bridge process (when $\\alpha\\rightarrow 0$).
> - We **derive the OU bridge mixture** from the OU bridge processes and present novel parameterization using the destination mixture (Eq. (5)).
> - We propose a **novel training objective (Eq. (9))** that can effectively estimate the destination mixture and further guarantees maximizing the likelihood.
>
> Additionally, we demonstrate that predicting the destination mixture enables faster convergence of the generative process achieved in the early stage as visualized in Figures 16~19 resulting in valid graphs and stable molecules, which was not studied in previous works.

---

> ### Author Response · Authors · 2023-11-15
> **Initial Response to Reviewer HfR4 (2/2)**
>
> **Comment 1-2:**
> Why is the framework related to graph topology?
>
> **Response 1-2:**
> Our framework is specialized for graph generation in threefold: (1) the motivation, (2) the parameterization and the training objective, and (3) the exploitation of graph structure.
>
> First, the motivation for our work is that the key to generating valid graphs is **modeling the discrete structures and the properties determined by the structure**, which previous objectives of graph diffusion models are ill-suited. Due to the discreteness and the combination of node features and adjacency, graph data is distinguished from other data such as images, and it is crucial to model the graph topology for the generation.
>
> From this motivation, our goal is to **directly learn the graph structure by estimating the weighted mean of data** (destination mixture), and we propose a novel parameterization of the OU bridge mixture using the destination mixture (Eq. (7)) and further derive novel training objective (Eq. (9)) for estimating the destination mixture.
>
> Moreover, to effectively estimate the destination mixture of graph data, **we exploit the discrete structure of graphs and generate both the node and adjacency simultaneously**. For modeling the discreteness, we utilize an additional function (e.g., sigmoid or softmax) at the end of the model, as explained at the end of Section 3.2 and Section 4.4. Further, we use the framework for attributed graphs (Section A.4) which describes the generative process of both the node features $\\textbf{X}$ and the adjacency matrices $\\textbf{A}$. These also contribute to the superior performance of DruM.
>
> ---
>
> **Comment 2:**
> How can Eq. (9) be used to estimate the destination mixture?
>
> **Response 2:**
> This is because **minimizing Eq. (9) is equivalent to minimizing Eq. (8)**, where Eq. (8) is designed so that the model $s\_{\\theta}$ to estimate $\\textbf{D}(\\textbf{G}\_t, t)$. As derived in Section A.6, the equivalence is due to the fact that
> - The expectation of Eq. (8) is computed over all the bridge processes in the mixture process (Eq. (47)).
> - The destination mixture is a weighted mean of data (Eq. (49)).
> Therefore, estimating the destination mixture can be achieved by minimizing Eq. (9).
>
> A similar case of equivalence in training objectives can be found in previous works, for example, Song et al. [3]: The denoising score matching, which aims to estimate the score function, is equivalent to the continuous-time score matching objective in Eq. (7) of [3] which minimizes the loss between the model output and the log gradient of the transition distribution. Although Eq. (7) of [3] seems to train the model for estimating the log gradient of the transition distribution, the equivalence guarantees that it actually trains the model to estimate the score function.
>
> [3] Song et al., Score-Based Generative Modeling through Stochastic Differential Equations, ICLR 2021
>
> ---
> ---
>
> We thank the reviewer again for their time and feedback, and we hope that our responses have addressed any remaining questions. We hope the reviewer would kindly consider a fresh evaluation of our work given our responses above and the revised paper.

---

> ### Author Response · Authors · 2023-11-21
> **Gentle Reminder for Reviewer HfR4**
>
> Dear Reviewer HfR4,
>
>
> Thank you for reviewing our paper. As the interactive discussion phase will end this Wednesday (22nd AOE), we politely ask you to check our responses. We summarize our responses as follows:
>
>
> - We made clear that the superior performance of our framework comes from our novel contribution in the parameterization and the training objective.
> - We clarified that our framework is specialized for generating graphs, from the motivation for modeling the topology to the parameterization and the training objective, as well as the exploitation of the graph structures.
> - We explained that our training objective of Eq. (9) is equivalent to Eq. (8) which is designed for estimating the destination mixture.
>
>
> We hope that our responses have addressed your concerns, and we hope you kindly consider updating the rating accordingly. Please let us know if there are any other things that we need to clarify or provide. We sincerely appreciate your valuable suggestions.
>
> Best regards,
>
> Authors

---

> > ### Comment · Reviewer_HfR4 · 2023-11-22
> > **Re-evaluation**
> >
> > Many thanks for the detailed response. I accept the claimed novelty as composed to the references [1][2]. Now, I understand the equivalence between (8) and (9). However, the connection to graph generation is not clear to me. The relation to graphs mentioned in the response sounds related to some implementation specifics, which is irrelevant to a mixture of the OU bridge process, the technical key ingredient of this work. Also, why the adopted method is good for graph generation is also unclear, though as claimed by the authors, the experiments have shown better empirical performance.
> >
> > Overall, I evaluate think work as a borderline work. I am okay with scores 5 or 6. Given the clear claimed novelty beyond [1,2], I increased my rating to 6.
> >
> > [1] Diffusion-based molecule generation with informative prior bridges, NeurIPS 2022
> > [2] Learning Diffusion Bridges on Constrained Domains, ICLR 2023

---

> > > ### Author Response · Authors · 2023-11-23
> > > **Thank you for your valuable feedback**
> > >
> > > We sincerely appreciate the reviewer’s time and effort in reviewing our paper. We are grateful for your valuable comments acknowledging the novelty of our work. We would like to address your remaining comments below:
> > >
> > > ---
> > >
> > > **Comment 3:**
> > > Why the adopted method is good for graph generation is unclear, though as claimed by the authors, the experiments have shown better empirical performance.
> > >
> > > **Response 3:**
> > > We would like to further clarify why our framework is especially beneficial for graph generation.
> > >
> > > A major difference between graph data and image data is that while a slight modification of the edges may significantly change the structure of the graphs and hence its properties, for example, connecting a pair of nodes can break the planarity or a change in the bond type of a molecule can transform the property from beneficial to toxic (e.g., Figure 1 of [1]), it is not the case for images where a slight change in the pixel values does not notably affect its semantic contents or its property (e.g., image class).
> > >
> > > Therefore, unlike image data, the key to generating valid graphs is **accurately modeling the discrete structures of graphs and the properties determined by the structure**. Generating graphs by estimating the noise (or score) is ill-suited since what the model learns is to denoise each step, not learning the graph structure. Since the topology is implicitly recovered from denoising, they cannot fully capture the structure.
> > >
> > > To address the limitation of previous works, **our model learns to predict the graph structure directly** through the destination mixture. By learning the accurate structure, **we can also capture the properties determined by the structure**, for example, planarity or clusteredness. Especially, when predicting the destination mixture, we can **exploit the discreteness of graphs** (e.g., the entries of the adjacency matrices are 0-1) by adding an additional function (e.g., sigmoid or softmax) at the end of our model, and as a result,  our model can easily capture the discrete structure.
> > >
> > > [1] Jo et al., Edge Representation Learning with Hypergraphs, NeurIPS 2021
> > >
> > > ---
> > >
> > > **Comment 4:**
> > > The relation to graphs mentioned in the response sound related to some implementation specifics, which is irrelevant to a mixture of the OU bridge process.
> > >
> > > **Response 4:**
> > > We would like to clarify that our **OU bridge mixture and our novel parameterization are essential components for achieving our goal of directly learning the graph structure through the destination mixture**. Since previous denoising diffusion models are sub-optimal for modeling the graph topology, we propose to predict the destination using the mixture of the OU bridge processes and our novel parameterization via the destination mixture.
> > >
> > > ---
> > > ---
> > >
> > > Thank you once again for reviewing our paper and thanks for your thoughtful comments.
> > >
> > > Best,
> > > Authors

---

### Official Review · Reviewer_w5ob · 2023-11-01

**Soundness:** 3 good
**Presentation:** 2 fair
**Contribution:** 4 excellent
**Rating:** 5
**Confidence:** 2

**Summary:**

This paper proposes Destination-Predicting Diffusion Mixture:
 - While DDPMs derive the generative process by reversing the forward noising process, DruM constructs the generative process from the mixture of OU bridge processes, which does not rely on the time-reversal approach. The mixture process of DruM defines an exact transport from an arbitrary prior distribution to the data distribution by construction.

**Strengths:**

Being a graph diffusion paper, the paper makes a very large contribution to the general diffusion literature, as well as the growing literature on diffusion bridge process. The paper, if read with the appendix, is well written. The experiment compares the proposed method to the most comparable recent works and demonstrates great performance.

**Weaknesses:**

- The paper mentions in numerous places that the proposed method captures the topology of the graph distribution because it predicts the destination of the diffusion process. I don't understand the arguments there, or if they are attempting to make one. x0-parameterization of DDPM (e.g. Vignac 2023) by definition predicts the destination. Does it capture topology? In fact, every generative model predicts destination in one way or another (VAE, GAN, and $\epsilon$-parameterized diffusion models are equivalent to x0-parameterization). Do they capture topology? "Loss-Guided Diffusion Models for Plug-and-Play Controllable Generation", for example, attempts to generate a destination "distribution" by doing MC on top of an x0-parameterization.
 - The paper assumes that the audience has prior knowledge on bridge processes. The only explanation offered is that they are "processes conditioned to an endpoint," which I find unhelpful. The writing can be improved in this regard.
 - The paper is without a discussion about the graph data structure in use; the omission is made explicit at the end of Sec 3. For this I find the writing difficult to follow at times. How does one take the weighted mean of discrete graphs and molecules? They are finally discussed in appendix B.3, but only in reference to other papers.
 - The main paper can really use an \Algorithm block from appendix B.2 (although not sure what to remove), to help the readers follow the sampling procedure.

**Questions:**

- See Weaknesses
 - Why graphs? It appears that the methodology proposed isn't at all particular to discrete/combinatorial data structure.

---

> ### Author Response · Authors · 2023-11-15
> **Initial Response to Reviewer w5ob (1/2)**
>
> We sincerely thank you for your constructive and helpful comments. We appreciate your positive comments that
> - Our work makes a very large contribution to the general diffusion literature as well as the literature on the diffusion bridge process.
> - Our method outperforms the most comparable recent works.
>
> We provide an updated revision of the paper reflecting your comments highlighted in orange, and initially address your concerns below.
>
> ---
>
> **Comment 1-1:**
> Do previous diffusion models ($\\epsilon$-parameterized, $x\_0$-parameterized) capture the graph topology? Every generative model predicts a destination in one way or another.
>
> **Response 1-1:**
> Previous objectives of graph diffusion models, learning the noise ($\\epsilon$) or $x\_0$, are **ill-suited for capturing the discrete structure of graphs.**
>
> Although every generative model predicts a destination in one way or another, $\\epsilon$-parameterized diffusion models fail to capture graph topology since they do not directly learn the graph structure from the dataset. Since these models learn to denoise each step to **implicitly** recover the topology, the predicted destination derived from the estimated noise (score) is sub-optimal for modeling the discreteness of the graph structure as well as the structural properties defined by the combination of node features and adjacency. We experimentally validate this by analyzing noise prediction models (ConGress and EDM) and score prediction model (GDSS):
>
> - Please check Figure 2 (Left), which shows that GDSS and ConGress fail to model the spectral topology, and, in Table 1, the high MMD results indicate that these models are unable to model the local graph topology.
>
> On the other hand, $x\_0$-parameterized (continuous) diffusion models, that predict the exact destination, are problematic for generating valid structure of graphs, since the **predictions are highly inaccurate in earlier steps**. Making an inaccurate prediction of the structure is critical since due to the discreteness of the graph structure, a small error in the prediction causes a very different topology. We experimentally validate this as follows:
>
> - Figure 8 (Right) shows that predicting the exact destination (EDM-Dest) results in lower molecule stability (35.95%) compared to ours (87.34%), and further shows lower results compared to the $\\epsilon$-parameterized model (EDM).
>
> Although $x\_0$ and $\\epsilon$-parameterization are mathematically equivalent, **what the model learns is completely different and crucially affects modeling the discreteness of the graph structure**. This is shown by the different performances of EDM and EDM-Dest.
> We can conclude that the objectives of previous graph diffusion models are sub-optimal for modeling the graph structure.
>
> ---
>
> **Comment 1-2:**
> How does the proposed framework capture the graph topology?
>
> **Response 1-2:**
> Instead of learning the noise ($\\epsilon$) or the exact destination ($x\_0$), our framework learns to **predict the weighted mean of the data** derived from the OU bridge mixture process (Eq. (4) and (5)), i.e., the destination mixture. In our work, we construct the generative process as the OU bridge mixture (Eq. (4)) where the process is driven toward the predicted destination mixture. By the definition of the destination mixture, it converges to the final graph to be generated, and therefore our generative process ends up in the correct structures within the data distribution. Based on the mixture process, we train our model to estimate the destination mixture which can directly learn the graph structure without the concern of making highly inaccurate predictions in the early steps. Therefore, our method can generate graphs with accurate topology, and we experimentally validate this as follows:
>
> - In Figure 2 (Left), DruM perfectly models the spectral topology, resulting in high validity (V.U.N.) compared to previous methods.
> - Figure 3 (Right) shows that DruM can capture the local topological properties of the graphs.
> - Visualization of the generation processes in Figures 16~19 indicates that our method can predict the topology at an early stage.

---

> ### Author Response · Authors · 2023-11-15
> **Initial Response to Reviewer w5ob (2/2)**
>
> **Comment 2:**
> It appears that the methodology proposed isn't at all particular to discrete/combinatorial data structure.
>
> **Response 2:**
> Our framework is designed and specialized for graph generation in threefold: (1) the motivation, (2) the parameterization and the training objective, and (3) the exploitation of graph structure.
>
> First, the motivation for our work is that the key to generating valid graphs is **modeling the discrete structures and the properties determined by the structure**, which previous objectives of graph diffusion models are ill-suited. Due to the discreteness and the combination of node features and adjacency, graph data is distinguished from other data such as images, and it is crucial to model the graph topology for the generation.
>
> From this motivation, our goal is to **directly learn the graph structure by estimating the weighted mean of data** (destination mixture), and we propose a novel parameterization of the OU bridge mixture using the destination mixture (Eq. (7)) and further derive novel training objective (Eq. (9)) for estimating the destination mixture.
>
> Moreover, to effectively estimate the destination mixture of graph data, **we exploit the discrete structure of graphs and generate both the node and adjacency simultaneously**. For modeling the discreteness, we utilize an additional function (e.g., sigmoid or softmax) at the end of the model, as explained at the end of Section 3.2 and Section 4.4. Further, we use the framework for attributed graphs (Section A.4) which describes the generative process of both the node features $\\textbf{X}$ and the adjacency matrices $\\textbf{A}$. These also contribute to the superior performance of DruM.
>
> However, as you mentioned, the extension of our work can be beneficial for generating other data structures where understanding the topology of the data is crucial.
>
> ---
>
> **Comment 3:**
> Detailed explanation of the bridge processes.
>
> **Response 3:**
> Due to the page limit, we provided the **detailed explanations in Appendix A.1**. Yet we agree that adding a detailed explanation would help the readers, and have updated the paper (Section 3.1) explaining that a bridge process can be obtained by conditioning a diffusion process, e.g., Brownian motion or OU process, to an endpoint using Doob’s h-transform.
>
> ---
>
> **Comment 4:**
> Explanation of the graph data structure.
>
> **Response 4:**
> A graph $\\textbf{G}$ with $N$ nodes is defined by a pair $(\\textbf{X}, \\textbf{A})$ where $\\textbf{X}\\in\\mathbb{R}^{N\\times F}$ is the node features with feature dimension $F$ and $\\textbf{A}\\in\\mathbb{R}^{N\\times N}$ is the weighted adjacency matrix. The weighted mean of the graph data can be represented by the pair of the weighted mean of node features and the weighted mean of the adjacency matrices. We have added an explanation of the graph data structure in the updated revision (beginning of Section 3).
>
> ---
>
> **Comment 5:**
> The main paper can use the pseudo-code in Appendix B.2 to help the readers follow the sampling procedure.
>
> **Response 5:**
> Although we have referenced Appendix B.2 in the main paper, we agree that moving the pseudo-code to the main paper would help the readers. Due to the page limit, we will move the pseudo-code in the final revision if we have additional pages. We have added a brief explanation of the sampling procedure in the revised paper (Section 3.2).
>
> ---
> ---
>
> We thank the reviewer again for their time and feedback, and we hope that our responses have addressed any remaining questions. We hope the reviewer would kindly consider a fresh evaluation of our work given our responses above and the revised paper.

---

> ### Author Response · Authors · 2023-11-21
> **Gentle Reminder for Reviewer w5ob**
>
> Dear Reviewer w5ob,
>
>
> Thank you for reviewing our paper. As the interactive discussion phase will end this Wednesday (22nd AOE), we politely ask you to check our responses. We summarize our responses as follows:
>
>
> - We clarified that previous diffusion models fail to capture the graph topology as their objectives are sub-optimal for accurately modeling the discrete graph structure and the properties determined by the structure.
> - We made clear that our framework is designed and specialized for the generation of graphs, including the motivation for modeling the topology, our novel parameterization and the training objective, and the exploitation of the graph structures.
> - We revised our paper following your suggestions by adding a detailed explanation of the bridge processes, the graph data structure, and the sampling procedure of our method.
>
>
> We hope that our responses have addressed your concerns, and we hope you kindly consider updating the rating accordingly. Please let us know if there are any other things that we need to clarify or provide. We sincerely appreciate your valuable comments.
>
> Best regards,
>
> Authors

---

> > ### Comment · Reviewer_w5ob · 2023-11-22
> > **Thank you for the response**
> >
> > Thank you for the response and the revised manuscript. All my questions were answered. Although I still think "capturing graph topology" is a very vague claim, I have a better understanding of what you are trying to convey, and will take it into account in my reevaluation.

---

> ### Author Response · Authors · 2023-11-22
> **Thank you for reviewing our paper**
>
> We sincerely appreciate the reviewer’s time and effort in reviewing our paper. We are happy to hear that all your concerns were addressed by our responses. Here, we would like to **clarify more about “capturing the graph topology” and why it is crucial for graph generation**.
>
> A **major difference between graph data and image data** is that while a slight modification of the edges may significantly change the structure of the graphs and hence its properties, for example, connecting a pair of nodes can break the planarity or a change in the bond type of a molecule can transform the property from beneficial to toxic (e.g., Figure 1 of [1]), it is not the case for images where a slight change in the pixel values does not notably affect its semantic contents or its property (e.g., image class).
>
> Therefore, unlike image data, the key to generating valid graphs is **accurately modeling the discrete structures of graphs and the properties determined by the structure** which corresponds to “capturing the graph topology”. As explained in *Response 1-1*, previous diffusion models fail to model the discreteness of the graph structures and hence the topological properties such as planarity or clusteredness. In contrast, our work can capture the graph topology by directly learning the graph structure via predicting the weighted mean of data (i.e., the destination mixture), as explained in *Response 1-2*.
>
>
> Thank you once again for reviewing our paper and providing valuable comments.
>
> [1] Jo et al., Edge Representation Learning with Hypergraphs, NeurIPS 2021
>
> Best,
> Authors

---

### Official Review · Reviewer_txBt · 2023-11-06

**Soundness:** 3 good
**Presentation:** 3 good
**Contribution:** 2 fair
**Rating:** 6
**Confidence:** 3

**Summary:**

This paper proposes to use score function to predict the final graph. The resulting stochastic process is named "Destination-Predicting Diffusion Mixture (DruM).

**Strengths:**

1. The method is theoretically sound. The underlying stochastic process is well defined.
2. The motivation to forecast the final state at the early de-noising stage is interesting.

**Weaknesses:**

1. The idea of "prediction results in graphs with correct topology" is not very convincing.
2. Empirical results are limited to small graphs.
3. The proposed method sacrifices novelty in generated graph.

**Questions:**

1. Does Table 4 illustrate a tradeoff between novelty and other capabilities for DruM? If this outcome is a direct consequence of your objective in Eq. (65), would it be advisable to introduce a control mechanism in your algorithm to balance these capabilities?
2. Is it your view that DruM could be better suited for generating large graphs, prioritizing validity and other capabilities over novelty? Do you believe DruM has the potential to alleviate the scalability bottleneck for larger graphs?

---

> ### Author Response · Authors · 2023-11-15
> **Initial Response to Reviewer txBt (1/2)**
>
> We sincerely thank you for your constructive comments. We appreciate your positive comments that
> - Our method is theoretically sound with a well-defined diffusion framework.
> - Motivation to predict the final state at the early stage is interesting.
>
> We provide an updated revision of the paper highlighted in orange, addressing all your concerns below.
>
> ---
>
> Before addressing your concern, we would like to **recap the main contributions of our work**:
> - We propose a new graph generative framework that models the graph topology by directly predicting the destination of the generative process as a weighted mean of data.
> - We derive theoretical groundwork for modeling the generative process using the OU bridge mixture and introduce a novel training objective for estimating the destination mixture.
> - Our framework has benefits that can directly predict the destination of the diffusion process while being able to handle potential continuous features.
> - Our method significantly outperforms previous graph generative models on diverse generation tasks.
>
> Now we initially address your concerns below.
>
> ---
>
> **Comment 1:**
> The idea of "prediction results in graphs with correct topology" is not very convincing.
>
> **Response 1:**
> Our method is superior in capturing the correct topology because **we directly learn the discrete structure of graphs by learning to predict the destination mixture** which is a weighted mean of data, in contrast to predicting noise or score function as done in previous works. In order to capture the topology, it is best for the model to directly learn the structure since the topology of a graph is its discrete structure and the corresponding properties of the structure.
>
> To this end, we aim to predict the graph structure as a weighted mean of graph data, namely the destination mixture $\\textbf{D}(\\textbf{G}\_t, t)$ (Eq. (1)), which converges to the final graph to be generated by its definition. In our work, we construct the generative process as the OU bridge mixture (Eq. (4)) where the process is driven toward the destination mixture, ending up in the correct structures within the data distribution. Thus we train our model to estimate $\\textbf{D}(\\textbf{G}\_t, t)$, and in this way, the prediction of $\\textbf{D}(\\textbf{G}\_t, t)$ from our framework results in a graph with accurate topology.
>
> We empirically validate this through extensive experiments:
>
> - Figure 2 (Left) shows that DruM perfectly models the spectral topology of the dataset through the diffusion process.
> - Figure 2 (Right) shows that DruM can capture the local topological properties of the graph.
> - Visualization of the generation processes in Figures 16~19 indicates that our method can generate accurate graph topology at an early stage.
>
> ---
>
> **Comment 2-1:**
> The proposed method sacrifices the novelty of the generated graphs.
>
> **Response 2-1:**
> This is a misunderstanding as our method does not sacrifice the novelty. Our **DruM is able to generate valid graphs that are different from that of the training set (novel)**, and the generated graphs are different from each other and are not limited to specific structures, which we experimentally validate as follows:
>
> - For general graph generation, we report the V.U.N. (valid, unique, and novel) metric in Table 1 where ours achieves the highest V.U.N. with 100% novelty. In other words, our method is able to generate valid graphs that are novel.
> - For molecule generation, we report the novelty as well as the uniqueness in Table 5 where ours achieves 99.98% novelty and 99.97% uniqueness for the ZINC250k dataset. Note that the low novelty in the QM9 dataset, similar to that of DiGress, is due to the fact that the molecules in the QM9 dataset only consist of a small number of atoms (up to 9 atoms) of four atom types (C, N, O, F).
>
> | DruM (ours)    | Planar | SBM   | Proteins | QM9   | ZINC250k |
> |----------------|--------|-------|----------|-------|----------|
> | Uniqueness (%) | 100.0  | 100.0 | 100.0    | 96.90 | 99.97    |
> | Novelty (%)    | 100.0  | 100.0 | 100.0    | 24.15 | 99.98    |
>
> ---
>
> **Comment 2-2:**
> Does Table 4 illustrate a tradeoff between novelty and other capabilities for DruM?
>
> **Response 2-2:**
> No, our method **does not have a tradeoff between generation quality (e.g., low MMD, high validity, low FCD) and novelty**. DruM can generate valid graphs (high validity, high V.U.N.) where the structural properties follow the data distribution (low MMD, low FCD, low NSPDK) and further not seen in the training dataset (high novelty), as explained in *Response 2-1*.
>
> In Table 4, the low novelty in QM9, similar to that of DiGress, is due to the fact that the molecules in the dataset consist of a small number of atoms (up to 9 atoms) of four atom types (C, N, O, F). Experiments on larger molecules (ZINC250k) verify that our method is able to generate novel molecules.

---

> ### Author Response · Authors · 2023-11-15
> **Initial Response to Reviewer txBt (2/2)**
>
> **Comment 3:**
> Empirical results are limited to small graphs
>
> **Response 3:**
> We respectfully disagree as we have experimentally evaluated on large real-world graph benchmark datasets, the Protein dataset (up to 500 nodes) and the GEOM-DRUGS dataset (up to 181 atoms with 16 atom types), that have been widely used in previous works. Our DruM significantly outperforms existing works on these large datasets achieving high validity.
>
> ---
>
> **Comment 4:**
> Do you believe DruM has the potential to alleviate the scalability bottleneck for larger graphs?
>
> **Response 4:**
> Although the goal of our framework does not lie in alleviating the scalability bottleneck, we empirically demonstrate that our method shows superior performance for generating large graphs (e.g., Proteins and GEOM-DRUGS) providing promising direction for the generation of very large graphs.
>
> ---
> ---
>
> We thank the reviewer again for their time and feedback, and we hope that our responses have addressed any remaining questions.

---

> ### Author Response · Authors · 2023-11-21
> **Gentle Reminder for Reviewer txBt**
>
> Dear Reviewer txBt,
>
>
> Thank you for reviewing our paper. As the interactive discussion phase will end this Wednesday (22nd AOE), we politely ask you to check our responses. Please let us know if there are any other things that we need to clarify or provide. We sincerely appreciate your constructive suggestions.
>
> Best regards,
>
> Authors

---

### Official Review · Reviewer_cip6 · 2023-11-17

**Soundness:** 4 excellent
**Presentation:** 3 good
**Contribution:** 3 good
**Rating:** 8
**Confidence:** 3

**Summary:**

This work outlines the destination-predicting diffusion mixture, which is a novel framework using a mixture of learned OU bridge processes to generate graph data. The purpose of this method, as opposed to traditional diffusion approaches which invert a learned a mapping from noise to data, is to explicitly predict the (distribution of the) destination of a diffusion process sending noisy samples to a data distribution. The main claim is that this destination prediction facilitates more effective modeling of graph-structured data, as opposed to typical diffusion models like DDPM. This claim is substantiated by extensive experiments on diverse datasets of small to large graphs.

**Strengths:**

1. The explanation of the modeling approach is fairly clear, though some of the notation is a bit confusing. Regardless, the authors do well to explain how the mixture of bridge processes is constructed, finally leading to a straightforward objective in Eq (9).
2. The experiments are very thorough; the GEOM-DRUGS result is particularly demonstrative of the improved performance of the destination prediction of DruM. The chemical/physical metrics of Table 2 and Figure 3 do well to highlight that this method is well-suited to real-data scenarios.
3. The simulation-free training and rapid convergence to the destination distribution is a significant advantage when compared to expensive SDE simulation in typical diffusion.

**Weaknesses:**

1. The main weakness in the paper is the lack of clarity about the argument behind the central claim that destination prediction facilitates more effective modeling of graph topological data. While the experimental results clearly show that this is the case, I failed to understand why this would be the case as I read the paper. The methodology described in Section 3 is completely agnostic to the fact that we are considering graph data, and while I can understand why it might be in general advantageous to predict the destination of diffusion, I'm not sure why this is particularly the case for graphs. The authors claim that destination prediction explicitly learns graph structure, but it is not clear to me why this is the case. Maybe this can be explained more clearly early in the text, towards the beginning of section 3.
2. The sampling procedure does not seem to be explained in the main text. Figure 1 (b) demonstrates the diffusion process at sampling time, but since this is not explained in the text, it is a bit confusing to see how to connect the learned ${\bf s}_\theta$ to the sampling procedure. It would be better to include Algorithms 1 and 2 in the main body of the text, including a brief explanation of Algorithm 2.

Despite my concerns above, I think this paper outlines a novel, effective procedure for graph generation. I believe some clarity in the text regarding the main claim and the sampling procedure would greatly enhance the clarity of the contribution.

**Questions:**

1. As above, how exactly does destination prediction relate to better graph generation? It is mentioned that it takes the graph topology into account more explicitly, but it is not clear to me how this is the case.
2. How does this approach compare to denoising-diffusion-type models in non-graph settings? While I understand this is not the focus of the paper, I think conducting a small experiment or at least commenting on this would make it clear why destination prediction is well-suited to graph data. As in the previous question, it is not clear to me why destination prediction is connected to the topology of the data.

---

> ### Author Response · Authors · 2023-11-19
> **Initial Response to Reviewer cip6 (1/2)**
>
> We sincerely thank you for your constructive and helpful comments. We appreciate your positive comments that
> - The paper outlines a novel, effective procedure for graph generation
> - The authors explain well how the bridge processes are constructed, finally leading to a straightforward objective in Eq (9).
> - The experiments are very thorough and chemical/physical metrics highlight that it is well-suited to real-data scenarios.
> - The simulation-free training and rapid convergence to the destination distribution is a significant advantage compared to previous diffusion models.
>
> We provide an updated revision of the paper highlighted in orange, and we initially address your concerns below:
>
> ---
>
> **Comment 1:**
> While the experimental results clearly show destination prediction facilitates more effective modeling of graph topological data, and while I can understand why it might be in general advantageous to predict the destination of diffusion, I'm not sure why this is particularly the case for graphs.
>
> **Response 1:**
> A major difference between graph data and image data is that while a slight modification of the edges may significantly change the structure of the graphs and hence its properties, for example, connecting a pair of nodes can break the planarity or a change in the bond type of a molecule can transform the property from beneficial to toxic (e.g., Figure 1 of [1]), it is not the case for images where a slight change in the pixel values does notably affect its semantic contents or its property (e.g., image class).
>
> Therefore, unlike image data, the key to generating valid graphs is **accurately modeling the discrete structures of graphs and the properties determined by the structure**. Generating graphs by estimating the noise (or score) is ill-suited since what the model learns is to denoise each step, not learning the graph structure. Since the topology is implicitly recovered from denoising, they cannot fully capture the structure as shown in Figure 2 (Left) where GDSS and ConGress fail to model the spectral topology.
>
> To address the limitation of previous works, **our model learns to predict the graph structure directly**. By learning the accurate structure, **we can also capture the properties determined by the structure**, for example, planarity or clusteredness. Specifically, by constructing the OU bridge mixture, we propose the destination mixture matching objective (Eq. (9)) where the graph structure is explicitly learned. Especially, in contrast to estimating the noise, we can **exploit the discreteness of graphs** (e.g., the entries of the adjacency matrices are 0-1) when predicting the structure by adding an additional function (e.g., sigmoid or softmax) at the end of our model. In this way, our model can easily capture the discrete structure, which we validate in Figure 2 (Right) achieving a significant decrease in the model complexity.
>
> We experimentally validate in Figure 2 (Left) and (Middle) that **our model predicts the accurate structure of graphs** (low MMDs) at the halfway stage of the generation and similarly observed in the visualization of the generation processes in Figures 16~19.
>
> We would like to emphasize that our framework is tailored for graph generation in the following aspects:
>
> - The motivation of our work is to capture the graph topology by predicting the destination.
> - This leads to our novel parameterization of the bridge mixture (Eq. (5)) through the destination mixture, and the training objective (Eq.(8) and (9)).
> - The parameterization and the training objective allow us to exploit the discrete graph structure for effectively learning the destination mixture (Section 3.2 paragraph: Advantages of our framework).
> - To generate attributed graphs (graphs with node features), we simultaneously generate both the node features and the adjacency matrices using the system of SDEs in Eq. (31) (Section A.4).
>
> All the components contribute to the superior performance of our DruM validated by extensive experiments.
>
> Following the reviewer’s suggestion, we have added more explanation in the revised paper (beginning of Section 3) regarding the importance of directly modeling the graph structure.
>
> [1] Jo et al., Edge Representation Learning with Hypergraphs, NeurIPS 2021

---

> ### Author Response · Authors · 2023-11-19
> **Initial Response to Reviewer cip6 (2/2)**
>
> **Comment 2:**
> It would be better to include Algorithms 1 and 2 in the main body of the text, including a brief explanation of Algorithm 2.
>
> **Response 2:**
> Due to the page limit, we provided a detailed explanation of the sampling procedure in Appendix B.4 and the pseudo-code in Algorithm 2. Yet we agree that adding a brief explanation would help the reader, and we have added an overview of sampling from DruM in Section 3.2 of the revised paper explaining:
>
> - By starting from samples from the prior distribution, we simulate the parameterized bridge process of Eq. (7) from time $t=0$ to $t=T$, where the drift is computed from the trained model $s_{\\theta}$.
> - We can leverage any SDE solver used in previous works, for example, the Euler-Maruyama method or the Heun's 2nd-order method.
> - Especially, we simultaneously generate both the node features ($X$) and adjacency matrices ($A$) by simulating the system of SDEs as in Eq. (31) (described in Section A.4).
>
> We would like to move the pseudo-code to the main paper in the final revision if we have additional pages.
>
> ---
>
> **Comment 3:**
> While I understand it is not the focus of the paper, how does this approach compare to denoising-diffusion-type models in non-graph settings?
>
> **Response 3:**
> In order to compare our approach with DDPM on image generation, we trained on the CIFAR-10 dataset following the training setup and the model architecture of [2]. We achieved an FID of 3.95 (until Nov. 19) which shows comparable results to DDPM (FID: 3.17) and slightly inferior to Score SDE (FID: 2.20).  Please note that we could not perform a proper hyperparameter search due to limited time. Yet we observe that our framework does not provide particular benefits for image generation. However, we believe our approach could be beneficial for generating data where understanding the topology of the data is crucial. We provide the samples of generated images in the following anonymized link: https://anonymous.4open.science/r/paper3082_rebuttal/sample.png.
>
> [2] Song et al., Score-Based Generative Modeling through Stochastic Differential Equations, ICLR 2021
>
> ---
> ---
>
> We thank the reviewer again for their time and feedback. We hope that our responses have addressed any remaining questions.

---

> > ### Comment · Reviewer_cip6 · 2023-11-19
> >
> > Thank you for the thorough response. The additions regarding the importance of leveraging structural properties of the data and regarding sampling do well to clarify the contribution of the work. I agree that, if space permits, moving the pseudo-code to the main paper in a final revision would be worthwhile, as well. I would say that adding some comment or brief results of the experiment you conducted on CIFAR-10, noting that your approach is most beneficial when applied to data with specific topological properties rather than typical image data, could also help clarify why your model is working well in the graph generation setting. Regardless, I think this work outlines a strong, novel approach, and that the added information in the text strengthens the clarity of the contribution. I've raised my score, accordingly.

---

> > > ### Author Response · Authors · 2023-11-20
> > > **Thank you for your valuable feedback**
> > >
> > > We sincerely appreciate the reviewer’s time and effort in reviewing our paper. We are grateful for your valuable feedback that helped enhance the presentation of our paper. We will move the pseudo-codes to the main paper if space permits, and further add the analysis on the image generation in the Appendix. Thank you once again for the thoughtful comments.
> > >
> > > Best,
> > > Authors

---

### Author Response · Authors · 2023-11-20
**Global Response by Authors**

Dear Reviewers,

We sincerely thank you for reviewing our paper and for the insightful comments and valuable feedback. We appreciate the positive comments that emphasize the novelty of our work and the advantages of our proposed method:

- DruM is a **novel diffusion-based graph generation framework** (cip6, HfR4) that makes a **very large contribution** to the general diffusion literature as well as the literature on the diffusion bridge process (w5ob).
- The framework is **theoretically sound** (txBt) and **well explained** (cip6, txBt).
- The proposed method can **directly predict the destination** of the diffusion process while being able to **handle potential continuous features** (HfR4).
- The **simulation-free training** and **rapid convergence** to the destination distribution is a significant advantage (cip6).
- The **experiments are very thorough** (cip6) and demonstrate the **effectiveness of the framework** (cip6, w5ob, HfR4).

Following the detailed comments from the reviewers, we have enhanced the presentation of our work by adding the following in the revised paper (highlighted in orange):

- We highlight the importance of directly modeling the graph structure.
- We briefly describe the sampling procedure of DruM in the main paper (originally in Appendix B.4 and Algorithm 2).
- We included a detailed explanation of the bridge process in the main paper (originally in Appendix A.1).
- We explain the graph data structure in the main paper (originally in Appendix A.4).

Thank you again for your thorough review and thoughtful suggestions. We hope our responses and the clarifications have addressed any remaining questions, and we are willing to address any further inquiries you may have.

Yours sincerely,

Authors

---

### Meta-Review · Area_Chair_86So · 2023-12-15

**Metareview:**

The paper presents the Destination-Predicting Diffusion Mixture (DruM), a new framework for generating graph data using learned OU bridge processes, aiming to improve the modeling of graph-structured data. This new approach, diverging from traditional diffusion models like DDPMs, demonstrates enhanced accuracy in capturing graph topology, as validated by experiments on various graph datasets. However, as pointed out by reviewers, there is a major drawback in that although the paper asserts that the proposed method captures the topology of the graph distribution, yet the connection to graph topology and the reasons for its effectiveness in graph generation remain unclear and requires further work.

**Justification For Why Not Higher Score:**

The paper has a major drawback in the lack of clarity of graph modeling.

**Justification For Why Not Lower Score:**

NA.

---

### Decision · Program_Chairs · 2024-01-16

Reject